# Early prediction of Alzheimer's disease using longitudinal electronic health records of US military veterans
Rumeng Li [1,2], Dan Berlowitz[1,3,4], Jesse Mez[5,6,7], Brian Silver[8], Xun Wang[9], Wen Hu[1,10], Raelene Goodwin[1,10], Heather Keating[1,10], Weisong Liu[1,4,10], Honghuang Lin [11] & Hong Yu [1,2,4,10] ✉

## Abstract

**Background** Early prediction of Alzheimer's disease is important for timely intervention and treatment. We examine whether machine learning on longitudinal electronic health record notes can improve early prediction of Alzheimer's disease.
**Methods** From Veterans Health Administration records (2000 to 2022), we studied 61,537 individuals diagnosed with Alzheimer's disease and 234,105 without, aged 45–103 years, 98.4% were male. From clinical notes, we quantified the frequency of subjective cognitive decline and Alzheimer's disease-related keywords, and applied statistical machine learning models to assess their ability to predict future diagnosis.
**Results** Here we show that Alzheimer's-related keywords (e.g., "concentration," "speaking"), occur more often in notes of individuals who later develop Alzheimer's disease than in controls. In the 15 years preceding diagnosis, cases demonstrate an exponential increase in keyword mentions (from 9.4 to 57.7 per year), whereas controls show a slower, linear increase (8.2 to 20.3). These trends are consistent across demographic subgroups. Random forest models using these keywords for prediction achieve an area under receiver operating characteristic curve from 0.577 at ten years before diagnosis to 0.861 one day before diagnosis, consistently outperforming models using only structured data.
**Conclusions** Signs and symptoms of early Alzheimer's disease are reported in clinical notes many years before a clinical diagnosis is made and the frequency of these signs and symptoms, approximated by keywords, increases the closer one is to the diagnosis. A simple keyword-based approach can capture these signals and can help identify individuals at high risk of future Alzheimer's disease.

## Plain language summary

This study explored whether early signs of Alzheimer's disease could be detected in routine medical records. We analyzed the health records of over 295,000 people from the U.S. Veterans Health Administration. We focused on words in doctors' notes that reflect a wide range of early symptoms, including changes in memory, speech, cognition, mood, physical functioning, and daily activity needs. These signs appeared more frequently and increased more rapidly in people who were later diagnosed with Alzheimer's. A computer model built on these words was able to predict who might develop the disease years in advance. These findings suggest that ordinary clinical notes could help doctors notice early warning signs of Alzheimer's and support earlier care and planning.

Alzheimer's disease (AD) is a progressive neurodegenerative disorder impairing memory and cognitive functions[1]. In 2024, an estimated 6.9 million people in the U.S. are living with AD, and this number is projected to rise to 13.8 million in the U.S. and over 100 million worldwide by 2060[2]. Early prediction of AD risk is crucial for effective interventions and

treatments, potentially enabling the development of therapies that slow or halt cognitive decline[1–4].

AD is characterized by amyloid plaques and tau tangles, which begin years before clinical symptoms appear. These pathological changes can be detected decades before dementia onset using biomarkers[2]. However, tests

[1]Center for Healthcare Organization & Implementation Research, VA Bedford Health Care System, Bedford, MA, USA. [2]Manning College of Information & Computer Sciences, University of Massachusetts Amherst, Amherst, MA, USA. [3]Department of Public Health, University of Massachusetts Lowell, Lowell, MA, USA. [4]Center of Biomedical and Health Research in Data Sciences, University of Massachusetts Lowell, Lowell, MA, USA. [5]Framingham Heart Study, Boston University Chobanian & Avedisian School of Medicine, Boston, MA, USA. [6]Alzheimer's Disease Research Center, Boston University Chobanian & Avedisian School of Medicine, Boston, MA, USA. [7]Department of Neurology, Boston University Chobanian & Avedisian School of Medicine, Boston, MA, USA. [8]Department of Neurology, UMass Chan Medical School, Worcester, MA, USA. [9]Microsoft Corporation, Redmond, WA, USA. [10]Miner School of Computer & Information Sciences, University of Massachusetts Lowell, Lowell, MA, USA. [11]Department of Medicine, UMass Chan Medical School, Worcester, MA, USA. ✉e-mail: hong_yu@uml.edu

like neuroimaging, cerebrospinal fluid analysis, and serum evaluations are often time-consuming, invasive, and expensive, limiting their widespread adoption in clinical practice[3,4].

Prior studies have leveraged patient speech records and other clinical data to detect AD. However, these studies have been limited in two main ways. First, they have focused on only one aspect of AD symptoms or risk factors, such as motor signs, spontaneous speech, blood pressure etc. rather than multiple ones[5–9]. Second, they have relied on diagnostic codes, medications, or procedure codes for psychological and cognitive testing, which are likely to be insensitive[10–18].

Subjective cognitive decline (SCD), wherein patients self-report cognitive difficulties before objective impairments are detectable through formal testing, is increasingly recognized as a precursor to future cognitive decline and AD dementia[19,20]. Its relevance is underscored by its inclusion in Stage 2 of Clinical staging for individuals on the AD continuum in the 2024 revised diagnostic and staging criteria for AD[21]. Although these early cognitive changes, along with associated physical and mental health alterations, may not be captured in diagnostic codes or structured data, they are often noticeable and frequently documented during routine healthcare visits, potentially reflected in the unstructured text of electronic health records (EHRs)[1,22]. Research has also shown that AD dementia related signs/symptoms can happen and be noticed long before its formal diagnosis. For example, in the Nun Study, linguistic density of writing in one's twenties predicted future AD risk[23]. Reports of memory loss and depressive symptoms occur up to 12 years prior to AD diagnosis[24]. Measures of accelerated long-term forgetting revealed cognitive decline in people with familial AD mutations up to a decade before the expected symptom onset[25,26]. Poor visual memory is associated with an increased risk of AD up to 15 years later[5]. These signs and symptoms are also potentially described in unstructured EHR notes and may be difficult to distinguish from normal aging[22]. Leveraging these clues may help early detection of AD risk[27]. However, manually reviewing years of EHR notes is costly and non-scalable.

To overcome this limitation, we analyze longitudinal electronic health records from more than 61,537 patients with AD and over 234,105 matched controls in the U.S. Veterans Health Administration. Using expert-curated keywords, we identify patterns of SCD and dementia-related signs and symptoms up to 15 years before diagnosis. We find that mentions of these terms increase gradually in controls but accelerate exponentially in patients who later develop AD, with a marked inflection in the final two years before diagnosis. Machine learning models trained on these narrative features predict Alzheimer's onset as early as ten years in advance, achieving substantially higher accuracy than models relying solely on structured EHR data. These results demonstrate that unstructured clinical text contains valuable early signals of Alzheimer's risk and can potentially enhance predictive modeling for earlier detection in real-world healthcare settings.

## Methods
### Setting and data sources
This retrospective cohort study used longitudinal EHR data from patients who received care through the U.S. Department of Veterans Affairs (VA) Veterans Health Administration (VHA) between October 1, 2000, and September 30, 2022. The VHA operates the largest integrated healthcare system in the United States, encompassing over 1200 facilities and providing a wide range of services, including inpatient, outpatient, mental health, rehabilitation, and long-term care. Nearly all medical activities are systematically documented in the VHA's Corporate Data Warehouse, which includes detailed information, such as diagnoses, procedures, prescriptions, and clinical notes, making it a valuable resource for large-scale, real-world research.

### Study design
Due to the high computational cost of analyzing longitudinal notes from a large patient cohort over a 20-year period, we conducted a case-control study instead of a cohort study. We identified AD cases using computable phenotypes (CPs), which define clinical conditions based entirely on EHR data without relying on manual chart review or clinician interpretation. Two CP strategies were employed to balance sensitivity and specificity.

To include all possible AD patients, we utilized the CP-I high-sensitivity approach. We identified AD cases within the VHA by selecting patients whose first AD diagnosis, based on ICD codes (Supplementary Table 1), occurred between October 1, 2015 (the implementation date of ICD-10 codes) and September 30, 2022 (the end of our study period). From the remaining VHA patients, we excluded those with any dementia diagnosis to create a control pool (Supplementary Table 2), matching each AD case with up to four controls based on age, sex, race/ethnicity, clinical utilization, Charlson comorbidity index (CCI), and area deprivation index (ADI) following previous work[18]. We considered the ADI as a surrogate for lifestyle and morbidity factors, reflecting socioeconomic conditions that can impact health outcomes. By incorporating the ADI, we aimed to account for the influence of environmental and social determinants of health on AD dementia, consistent with similar studies[28]. Fig. 1 outlines the inclusion and exclusion criteria used to generate the study cohort.

To enhance the specificity of AD case ascertainment in our CP-II cohort, we require that each AD case be diagnosed at least twice at separate time points. At least one of these diagnoses must be made at a specialty clinic—such as neurology, geriatrics, GeriPACT (Geriatric Patient Aligned Care Team), mental health, psychology, psychiatry, or geriatric psychiatry—by a provider specializing in neurology, vascular neurology, psychiatry, neuropsychology, or geriatric medicine. The clinic types are identified by Stop Codes (Supplementary Table 3) which are used by VHA to specify the type of outpatient care and workload of a visit[29]. The CP-II cohort was derived from the CP-I cohort by selecting AD cases that met these criteria, along with their matched controls.

Note that here the AD diagnosis is based on ICD codes that specify AD without explicitly mentioning dementia, it is important to acknowledge that in routine clinical practice, AD diagnoses are predominantly grounded in observable clinical symptoms, such as amnestic dementia, without necessarily relying on biomarker confirmation of underlying neuropathology. Consequently, our study is designed to forecast the clinical syndrome associated with AD as documented in medical records.

To build predictive models for AD onset, we determined an index time to identify input features prior to the first clinical indication of AD. For patients with AD, the index time was defined as the date of their first ICD-based AD diagnosis. For control patients, the index time was set as one year before their last recorded visit date. To ensure a consistent patient population and sufficient predictors for model training and evaluation, only patients who had over five years of EHR data prior to the index time were included in both the AD cases and control cohorts. We included patients younger than 65 years old since early-onset AD accounts for about 5–6% of cases[30]. By design, each control participant was matched to a single case participant, and case participants before diagnosis were not included as controls prior to their diagnosis.

The EHR consists of many different note types. In this study to manage computational resources while examining patients over a 20-year period within a large cohort, we focused on EHR notes generated from primary care visits, emergency visits, home-based primary care (HBPC), memory clinics, neurology, neuropsychology, geriatrics, psychiatry, psychology, cognitive care nursing, mental health clinics, compensation and pension examinations, and consultation visits.

### SCD and AD dementia-related keywords extraction & prevalence pattern analysis
We analyzed the longitudinal EHR notes of patients who have been diagnosed with AD and controls, to investigate the patterns and trends of SCD and AD dementia signs and symptoms in their clinical documentation. Building on an extensive literature review[22,31–35], and referencing existing expert-curated keyword approaches[36], we curated a list of 122 SCD- and AD dementia–relevant keywords through a structured, expert-driven process. Six domain experts contributed to the development of this list: three health

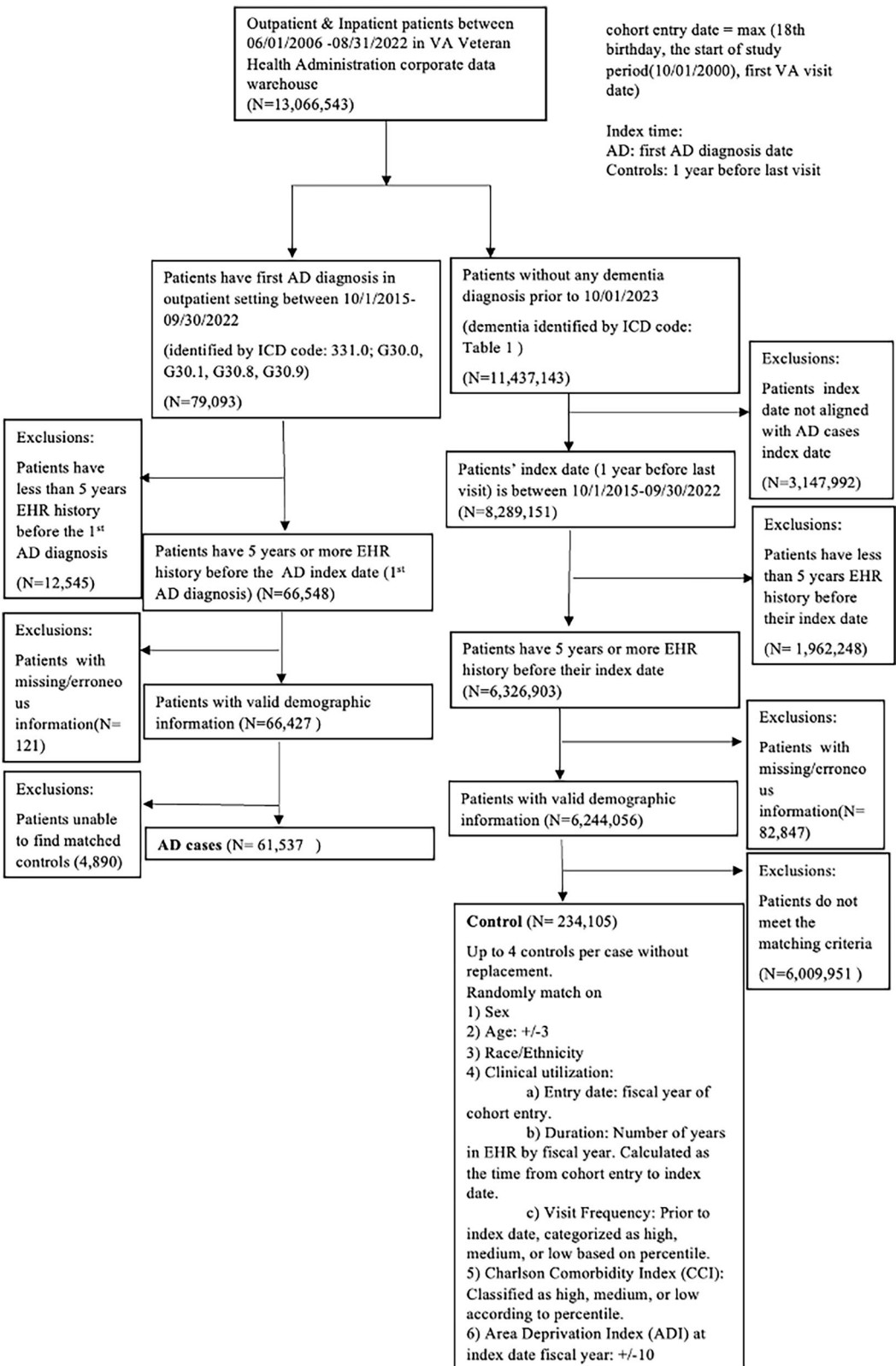

**Fig. 1 | AD case-control cohort creation flowchart.** This flowchart outlines the creation of the Alzheimer's disease AD case–control cohort from VHA EHRs. AD cases were identified using ICD-9/10 codes and required at least 5 years of EHR history prior to diagnosis. Each case was matched to up to four dementia-free controls on demographic and clinical factors. Exclusion criteria included insufficient EHR history, missing or erroneous demographic data, and inability to find matched controls. AD, Alzheimer's disease; EHR, electronic health record; VA, Veterans Affairs; ICD, International Classification of Diseases.

**Fig. 2 | Method overview: keyword extraction from clinical notes to AD prediction for decision support.** Workflow showing extraction of AD–related keywords from longitudinal electronic health records, which are converted into patient-level predictors for machine learning models. The framed section highlights the predictors and their contributions to the prediction. AD, Alzheimer's disease.

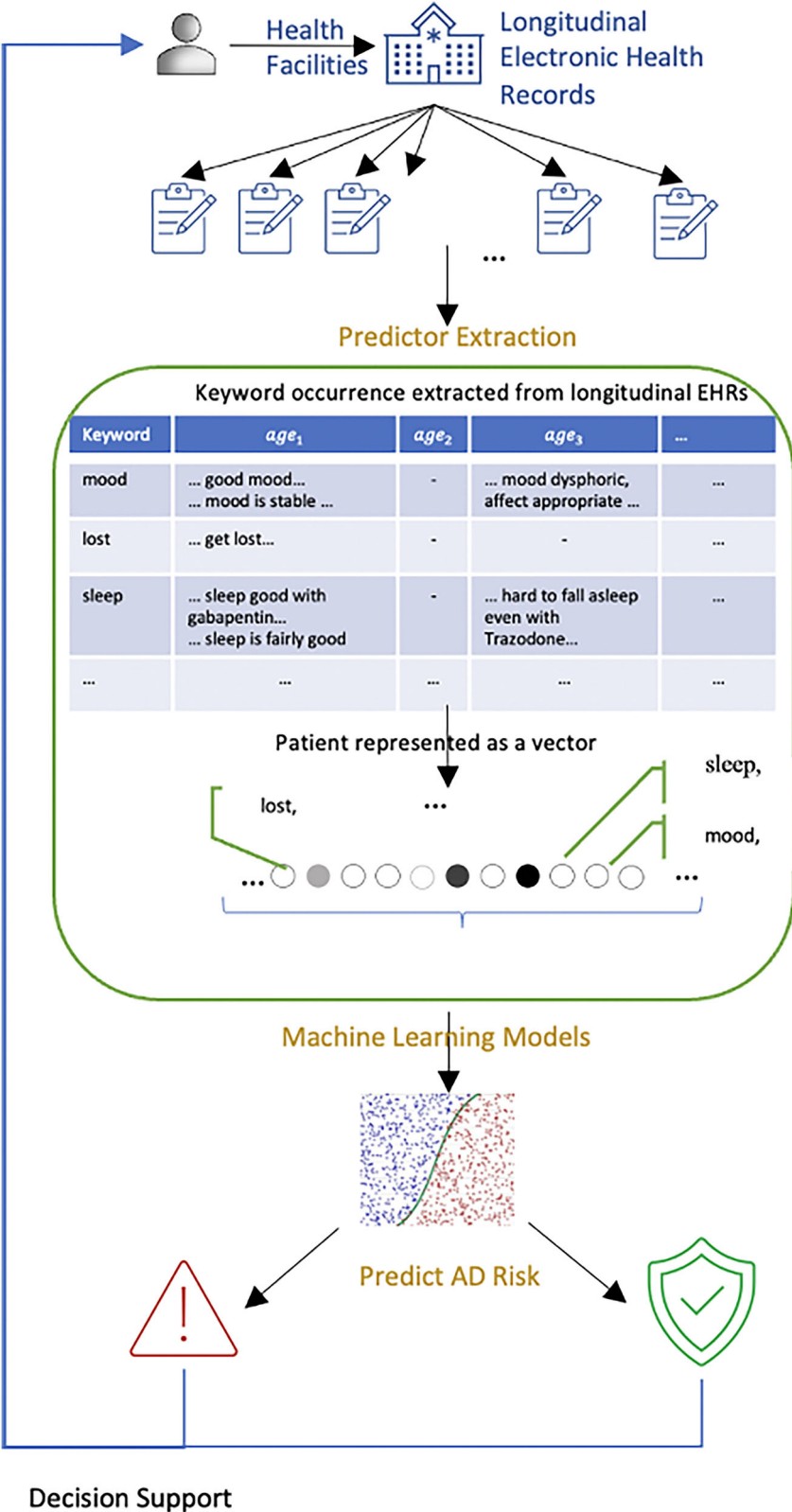

specialists with expertise in cognitive aging and EHR-based research, two neurologists, and one epidemiologist. The health specialists independently proposed candidate terms based on their clinical experience reviewing narrative EHRs and insights from a targeted literature review. The neurologists and epidemiologist provided clinical oversight through weekly meetings, helping validate the accuracy, consistency, and clinical relevance of the terms. The final list was refined through collaborative discussion and mutual agreement. Keywords were then grouped into clinically meaningful categories, including cognition–speech/language, cognition–memory, cognition–learning/perception, assistance needed, physiological changes, and neuropsychiatric symptoms (Supplementary Box 1). For each patient we identified the occurrence of keywords in longitudinal EHRs and tracked

their occurrences over time by aligning them to patients' age or year prior to diagnosis. Figure 2 provides illustrative examples on the keyword occurrences extraction. The patient's records state "good mood" at $age_1$ and "mood dysphoric" at $age_3$; the keyword "mood" is extracted for both without differentiating the contextual meanings. The keywords reflect clinicians' noting/documenting a feature of the patient. If the feature is concerning, it may be noted more frequently.

We examined the average yearly frequency of keywords approximating AD dementia–related signs and symptoms for up to 15 years before diagnosis. We analyzed all selected note types and further stratified the analysis by clinical specialty group to examine the symptom documentation patterns. In addition, we expected that patients with AD symptoms would have more frequent physician visit than those without such symptoms, and thus the AD cases would have more notes with AD-related keywords than the controls. To account for this potential bias, we normalized the AD keyword frequency by notes number. To examine the effects of biological variables on the patterns of AD signs and symptoms, we stratified the cohort by sex, age at diagnosis, and race/ethnicity. Moreover, we analyzed the patterns for each category of keywords, which correspond to different aspects of AD dementia signs and symptoms.

### Predictors extraction
Predictors studied in this work are the occurrence of AD dementia-relevant keywords in patients' EHR notes before the index time. We used the term frequency-inverse document frequency (TF-IDF) to identify the importance of each keyword. By using TF-IDF, we considered both the occurrence of a keyword and also the frequency with which it appears over time. Details for calculating the TF-IDF value are in Supplementary Note 1.

### Observation window and prediction window
For each setting, the observation window spanned from the start of the study period on 10/1/2000 to the start of a clean period in which no data was used in predicting AD. The length of this clean window could vary from 0 to 10 years (Supplementary Fig. 1). Those with less than 5 years of EHR history within the observation window were excluded. Keyword predictors were extracted from the observation window. When the observation window concluded at the AD index date (−1-day prior prediction), we included all the patients in our cohort for experiments. As the prediction period (clean window) was extended from 0 to 10 years, the number of people included in the experiments decreased.

### Machine learning models for predicting AD
Prediction of a patient at risk of AD can be formulated as a binary classification task. The inputs to the machine learning models are the predictors of a patient (vector representation) and the outputs are the risk of AD. We deployed machine learning models widely used for predictions in the clinical domain[37], mainly random forests (RF) model[38].

We selected RF as the primary classification model based on its strong balance between interpretability and predictive flexibility. RF models offer greater transparency than neural networks while capturing nonlinear relationships more effectively than logistic regression—an advantage particularly valuable in clinical contexts where understanding feature contributions is critical[18]. We also evaluated logistic regression[39] and XGBoost[40] as baseline models. Details of the models and model implementations are shown in Supplementary Note 1.

### Experiments setting
We conducted experiments to evaluate our models in predicting AD onset under two settings.

### Setting I
Random split. We randomly partitioned the study cohort into 70% training, 10% validation (for hyperparameter tuning and model selection), and 20% testing.

### Setting II
Hold- out stations. We aimed to evaluate the models' generalizability across different medical centers. To achieve this, we split the data by VHA stations, which are specific locations where veterans receive medical care. We randomly selected 13 out of 130 stations for testing, 13 for tuning, and the rest for training. Patients with records at multiple stations were assigned to their most frequently visited station.

### Statistics and Reproducibility
All analyses were performed at the patient level. Model discrimination was assessed using the area under the receiver operating characteristic curve (AUROC) and area under the precision-recall curve (AUPRC), with differences between AUROCs evaluated by DeLong's test (*pROC* package). Keyword lists are provided in the Supplementary. Source data for Figs. 3 and 4 are provided in Supplementary Data 1. All preprocessing steps, feature definitions (including TF–IDF and structured-feature mappings), and model hyperparameters are fully documented in Supplementary Note 1 to ensure reproducibility.

### Ethical approval
Our study protocol was approved by the institutional review board of US Veterans Affairs Bedford Health Care, and we obtained a waiver of informed consent due to minimal risk to participants.

## Results
### Sample characteristics
The CP-I cohort includes 61,537 AD cases (1,143 [1.9%] women; mean [SD] age at AD onset, 79.9 [8.7] years) and 234,105 controls (3486 [1.5%] women; mean [SD] age at index date, 79.6 [8.8] years), with an overall AD prevalence

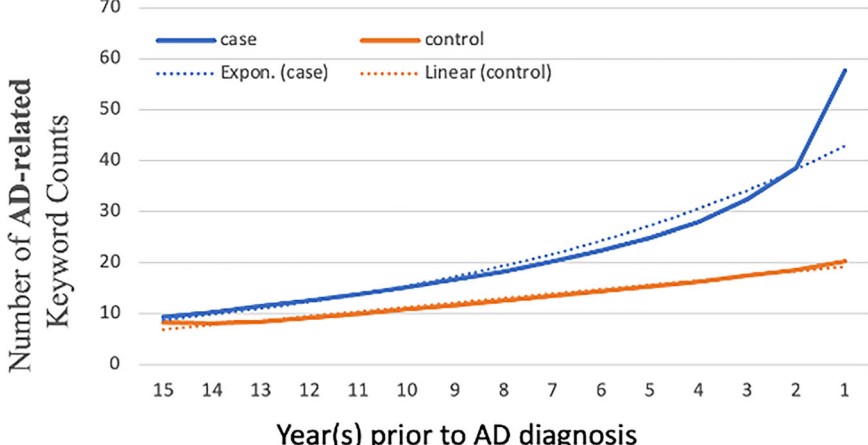

**Fig. 3 | SCD and AD dementia-related keyword patterns in longitudinal EHRs.** Average counts of SCD and AD dementia-related keywords per patient by year before diagnosis for AD cases and matched controls. Solid lines show observed data, and dotted lines show fitted trendlines (exponential for cases, linear for controls). Data are from the CP-I high-sensitivity computable phenotype cohort (AD cases: 61,537; controls: 234,105). Expert-curated SCD and AD dementia-related keywords were used to identify signs and symptoms. AD, Alzheimer's disease; SCD, subjective cognitive decline.

**Fig. 4 | SCD and AD dementia-related keyword patterns in longitudinal EHRs normalized by note count.** Average number of expert-curated AD and SCD–related keywords per clinical note in AD cases (*n* = 61,537) and matched controls (*n* = 234,105), plotted by year before AD diagnosis to account for clinical utilization. Keywords were identified from the CP-I high-sensitivity computable phenotype cohort. AD, Alzheimer's disease; SCD, subjective cognitive decline.

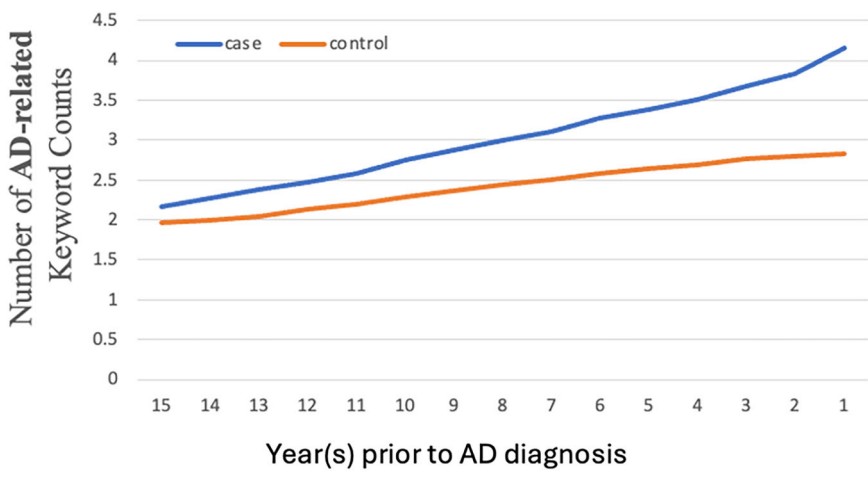

**Table 1 | Demographics of the CP-I cohort with AD cases having at least one ICD-based diagnosis**

| Characteristic | AD Cases | Controls |
|---|---|---|
| Patients, No. | 61,537 | 234,105 |
| Birth year, mean (s.d.) | 1937.8 (8.9) | 1938.4 (9.1) |
| AD onset/index time age (s.d.) | 79.9 (8.7) | 79.6 (8.8) |
| Sex, n. (%) | | |
| Female | 1143 (1.9%) | 3486 (1.5%) |
| Male | 60,394 (98.1%) | 230,619 (98.5%) |
| Race No. (%) | | |
| White | 49,537 (80.5%) | 192,430 (82.2%) |
| Black or African American | 6839 (11.1%) | 24,217 (10.3%) |
| Asian | 275 (0.4%) | 749 (0.3 %) |
| Native Hawaiian or Other Pacific Islander | 326 (0.5 %) | 942 (0.4%) |
| American Indian or Alaska Native | 175 (0.3%) | 493 (0.2%) |
| Unknown | 4385 (7.1%) | 15,274 (6.5 %) |
| Ethnicity No. (%) | | |
| Hispanic/Latino | 3345 (5.4%) | 10,249 (4.4 %) |
| Non- Hispanic/ Latino | 56,229 (91.4 %) | 217,091 (92.7 %) |
| Others/Unknown | 1,963 (3.2%) | 6765 (2.9%) |
| Number of notes with keywords (average) | 62.4 | 43.2 |
| Keywords of Interests number (average) | 308.5 | 175.8 |

*AD* Alzheimer's disease; *ICD* International Classification of Diseases; *CP-I* high-sensitivity computable phenotype.

**Table 2 | Demographics of the CP-II cohort with AD cases having at least two AD diagnosis code, one of which is from specialty clinics**

| Characteristic | AD Case | Control |
|---|---|---|
| Patients, No. | 35,308 | 145,198 |
| Birth year, mean (s.d.) | 1938.7 (8.7) | 1939.3 (8.9) |
| AD onset/index time age(s.d.) | 79.0 (8.4) | 78.6 (8.6) |
| Sex, n. (%) | | |
| Female | 705 (2.0 %) | 2150 (1.5 %) |
| Male | 34,333 (97.2%) | 143,048 (98.5%) |
| Race No. (%) | | |
| White | 27,868 (78.9%) | 119,195 (82.1 %) |
| Black or African American | 4547 (12.9%) | 15,121 (10.4%) |
| Asian | 182 (0.5%) | 467 (0.3%) |
| Native Hawaiian or Other Pacific Islander | 193 (0.5%) | 599 (0.4 %) |
| American Indian or Alaska Native | 98 (0.3%) | 316 (0.2%) |
| Unknown | 2150 (6.1%) | 9500 (6.5%) |
| Ethnicity No. (%) | | |
| Hispanic/Latino | 2314 (6.6%) | 10,249 (7.1 %) |
| Non- Hispanic/Latino | 31,867 (90.3%) | 131,447 (90.5%) |
| Others/Unknown | 857 (2.4%) | 3502 (2.4%) |
| Number of notes with keywords (average) | 67.5 | 47.1 |
| Keywords of Interests number (average) | 338.2 | 192.6 |

*AD*, Alzheimer's disease; *ICD*, International Classification of Diseases; *CP-II*, high-specificity computable phenotype.

of 20.8%. Detailed demographic and clinical characteristics are shown in Table 1.

The CP-II cohort was derived from the CP-I cohort and consists of 35,308 AD cases (705 [2.0%] women; mean [SD] age at AD onset, 79.0 [8.4] years) and 145,198 matched controls (2,150 [1.5%] women; mean [SD] age at index date, 78.6 [8.6] years), with an overall AD prevalence of 19.6%. Detailed characteristics of the CP-II cohort are shown in Table 2.

**Keywords related to SCD and AD dementia are more frequently mentioned in longitudinal EHRs of AD cases compared to controls**

Examining the CP-I cohort, we found more frequent keyword mentions in AD cases than controls. AD patients had an average of 308.5 keywords from 62.4 notes, spanning 14.7 years of their EHR history compared with controls who had 175.8 relevant keywords from 43.2 notes, spanning 13.5 years. We calculated the average number of mentions for each keyword group (Supplementary Fig. 2). Overall, AD cases consistently show higher mentions across all keyword groups compared to controls, indicating a greater prevalence of these issues in AD cases. Physiological changes and neuropsychiatric symptoms are the most frequently mentioned categories for both AD cases and controls. Consistent with established findings[22], AD cases in our study had almost six times the average mentions of cognition-memory issues compared to controls, underscoring the significant memory challenges faced by AD patients.

### Keywords related to SCD and AD dementia show accelerated increase patterns in the EHRs of AD patients 15 years leading up to diagnosis

Using data from the CP-I cohort, we found that within 15 years leading up to the first AD diagnosis, the number of keywords related to SCD and AD dementia in the clinical notes of AD patients increased exponentially—from 9.4 to 57.7 keywords per patient per year. In contrast, during the same period, the number of such keywords in the notes of control patients increased linearly, from 8.2 to 20.3 (Fig. 3). This exponential increase in AD patients was also evidenced when examining only primary care notes (Supplementary Fig. 3). In addition, we expected that patients with AD dementia symptoms or SCD would have more frequent physician visit than those without such symptoms, and thus the AD cases would have more notes with SCD and AD dementia-related keywords than the controls. To account for this potential bias, we normalized keyword frequency by the note number, and it showed similar patterns (Fig. 4). The pattern remained consistent across different ages at diagnosis (Supplementary Fig. 4) and showed little variation when stratifying the patient cohort by sex (Supplementary Fig. 5) and ethnicity/race (Supplementary Fig. 6).

When examining the patterns for keyword categories, the results indicated that AD cases exhibited a more rapid increase in SCD and AD dementia-related keyword groups compared to controls across all categories (Supplementary Fig. 7). However, the difference between the groups was more pronounced for some groups than others. For groups, such as cognition-memory, cognition-learning/perception, and neuropsychiatric symptoms, the cases had a sharp increase, while the controls remained relatively stable. For groups, such as physiologic changes, cognition-speech/language, and assistance needed, both cases and controls had a noticeable increase, but the cases still had a much higher rate of increase than the controls.

Similar patterns were observed in the CP-II cohort. The number of SCD and AD dementia-related keywords in clinical notes for AD patients increased exponentially from 10.0 to 65.1 per patient per year, while for control patients, the increase was linear, from 8.7 to 22.0 (Supplementary Fig. 8).

To better understand how symptom patterns manifest across different care settings, we compared the keyword frequencies across grouped note types. Due to data sparsity in certain categories, we aggregated notes into broader, clinically meaningful groups to ensure statistical reliability: Primary; Emergent Care (emergency visits); Mental Health (psychiatry, psychology, mental health clinics); Cognitive Specialty (memory clinics, neurology, neuropsychology, cognitive care); Geriatric Services (geriatrics, HBPC); and Consultation Services (consults, compensation & pension examinations). As shown in Supplementary Fig. 9, all note types exhibited increasing keyword trends among AD cases in the years preceding diagnosis. Mental health and geriatric service notes showed the highest average keyword counts and the steepest increases in the final 2–3 years, suggesting their key role in documenting late-stage symptom emergence. In contrast, primary care notes displayed a relatively high baseline and gradual increase beginning as early as 15 years prior to diagnosis (Supplementary Fig. 3), underscoring their importance in capturing early, longitudinal indicators. These findings highlight the complementary roles of generalist and specialist care settings in the documentation of prodromal AD symptoms. We also included the distribution of note types by specialty in the case and control cohorts in CP-I cohort in Supplementary Fig. 10.

Across all plots, an inflection point was observed around two years prior to diagnosis, prompting additional analyses of both SCD and AD dementia-related keyword frequencies and clinical note volume across Years 3, 2, and 1. As shown in Supplementary Fig. 11, all symptom categories exhibited increasing trends, with neuropsychiatric symptoms showing the steepest rise between Year 2 and Year 1—indicating a sharp escalation in behavioral and psychological symptom documentation. Physiological changes also increased during this period, while learning/perception and memory-related terms showed smaller but consistent gains. We further analyzed note volume by specialty (Supplementary Fig. 12) and

**Table 3 | Random forest prediction results using structured data features in Setting I on the CP-I cohort**

| Prediction Horizon | AUROC | AUPRC |
|---|---|---|
| −1 day | 0.682 | 0.571 |
| −1 year | 0.637 | 0.523 |
| −2 year | 0.601 | 0.487 |
| −3 year | 0.580 | 0.466 |
| −5 year | 0.552 | 0.425 |
| −7 year | 0.516 | 0.397 |
| −10 year | 0.497 | 0.371 |

*AUROC* Area Under the Receiver Operating Characteristic Curve; *AUPRC* Area Under the Precision–Recall Curve; *CP-I* high-sensitivity computable phenotype.

found that geriatrics and cognitive care clinics showed the largest relative increases between Year 2 and Year 1, reflecting intensifying care needs and diagnostic attention. These findings highlight the clinical significance of the Year 2 inflection point and the parallel escalation in both symptom documentation and healthcare utilization.

To assess the specificity of our symptom-based findings, we conducted an additional analysis using a curated set of dermatologic-related control keywords—terms commonly documented in clinical practice but not associated with AD dementia. The list included general symptoms (e.g., itching, rash), localized findings (e.g., skin lesion, ulcer), and dermatologic diagnoses (e.g., eczema, cellulitis), selected through a targeted review of clinical sources and screened for clinical relevance, EHR frequency, and independence from AD-related pathology (Supplementary Box 2). As shown in Supplementary Fig. 13, the average number of dermatologic-related keywords, normalized by note count, remained largely similar between AD cases and controls over time. This supports the interpretation that the observed temporal patterns in AD dementia-relevant keywords are not driven by general documentation trends or nonspecific symptom reporting.

### Machine learning models based solely on SCD and AD dementia-relevant keywords from EHR notes can predict AD onset up to 10 years in advance

Existing work primarily utilized structured EHR data like medications, ICD codes, and abnormal lab measurements for AD prediction[18], we evaluated the predictive performance of a random forest model using three distinct feature sets: (1) structured clinical features only (i.e., medications, ICD codes, and abnormal lab measurements), (2) keyword-derived features extracted from unstructured clinical notes, and (3) a combination of structured and keyword-based features.

The RF models in Setting I under random split produced the following results. Using only structured EHR features, the model achieved an AUROC of 0.497 at −10 years and 0.682 at −1 day, with corresponding AUPRCs of 0.371 and 0.571 on the CP-I cohort (Table 3). When using only keyword-derived features, the model substantially outperformed the structured-data-only approach, achieving an AUROC of 0.577 at −10 years and 0.861 at −1 day, and AUPRCs of 0.373 and 0.741 on the CP-I cohort (Table 4). Similar improvements were observed in the CP-II cohort, where AUROCs reached 0.598 at −10 years and 0.896 at −1 day. Combining structured features with keyword-derived features provided modest additional gains, with AUROCs increasing to 0.581 at −10 years and 0.867 at −1 day, and AUPRCs to 0.376 and 0.747 on the CP-I cohort (Table 5).

These results highlight that narrative symptom evidence extracted from unstructured clinical notes offers significant predictive value for early AD detection, even when used independently of traditional structured EHR fields. In addition to reporting results at the boundary time points of 10 years and 1 day before diagnosis, we also evaluated model performance at intermediate intervals of −1 year, −2 years, −3 years, −5 years, and −7 years prior to diagnosis, as detailed in Tables 3–5.

**Table 4 | Random forest prediction results using keyword features in Setting I**

| Prediction Horizon | CP-I | | CP-II | |
|---|---|---|---|---|
| | AUROC | AUPRC | AUROC | AUPRC |
| −1 day | 0.861 | 0.741 | 0.896 | 0.785 |
| −1 year | 0.772 | 0.603 | 0.781 | 0.609 |
| −2 year | 0.728 | 0.526 | 0.732 | 0.528 |
| −3 year | 0.705 | 0.488 | 0.707 | 0.476 |
| −5 year | 0.663 | 0.417 | 0.662 | 0.400 |
| −7 year | 0.628 | 0.369 | 0.637 | 0.358 |
| −10 year | 0.577 | 0.373 | 0.598 | 0.362 |

*AUROC* Area Under the Receiver Operating Characteristic Curve; *AUPRC* Area Under the Precision–Recall Curve; *CP-I* high-sensitivity computable phenotype; *CP-II* high-specificity computable phenotype.

**Table 5 | Random forest prediction results using a combination of structured data and keywords as features in Setting I on the CP-I cohort**

| Prediction Horizon | AUROC | AUPRC |
|---|---|---|
| −1 day | 0.867 | 0.747 |
| −1 year | 0.779 | 0.612 |
| −2 year | 0.734 | 0.532 |
| −3 year | 0.709 | 0.493 |
| −5 year | 0.666 | 0.417 |
| −7 year | 0.632 | 0.371 |
| −10 year | 0.581 | 0.376 |

*AUROC* Area Under the Receiver Operating Characteristic Curve; *AUPRC* Area Under the Precision–Recall Curve; *CP-I* high-sensitivity computable phenotype.

**Table 6 | Random Forest prediction results using keywords from primary care notes only and using Setting II (hold-out station 13/130) on the CP-I cohort**

| Prediction Horizon | Primary care notes only | | Setting II | |
|---|---|---|---|---|
| | AUROC | AUPRC | AUROC | AUPRC |
| −1 day | 0.810 | 0.644 | 0.862 | 0.738 |
| −1 year | 0.742 | 0.543 | 0.771 | 0.604 |
| −2 year | 0.713 | 0.474 | 0.726 | 0.527 |
| −3 year | 0.683 | 0.444 | 0.698 | 0.485 |
| −5 year | 0.644 | 0.384 | 0.651 | 0.406 |
| −7 year | 0.617 | 0.345 | 0.620 | 0.359 |
| −10 year | 0.569 | 0.354 | 0.577 | 0.373 |

*AUROC* Area Under the Receiver Operating Characteristic Curve; *AUPRC* Area Under the Precision–Recall Curve; *CP-I* high-sensitivity computable phenotype.

In identifying the informative prediction features, for models using keywords only features on both CP-I and CP-II cohorts, physiological changes and neuropsychiatric symptoms are generally the most important predictors at earlier prediction time points. In contrast, memory and cognitive learning/perception become more significant as the diagnosis approaches Supplementary Fig. 14. Top features across each time point model in the CP1 cohort included activities of daily living (ADLs), memory, executive function, instrumental activities of daily living (iADLs), attention, anxiety, pain, hearing, comprehension, and communication. The CP2 cohort had similar top decision features (Supplementary Fig. 15, 16).

We also trained models using only primary care notes to evaluate the effectiveness of data from general practitioners, yielding AUROC values from 0.569 (-10 years) to 0.810 (-1 day) and AUPRC values from 0.354 to 0.644 on the CP-I cohort (Table 6).

The RF models in Setting II of hold-out stations performed similarly to Setting I on the CP-I cohort, with AUROC values from 0.577 (-10 years) to 0.862 (-1 day) and AUPRC values from 0.373 to 0.738. This suggests the models maintained predictive ability across different medical centers (Table 6).

Supplementary Table 4 presents the performance of our baseline models. The RF model consistently outperformed logistic regression and achieved performance comparable to XGBoost. However, XGBoost was less interpretable when applied to sparse keyword features[41].

Supplementary Table 5 presents the random forest prediction results using keyword features from different specialty note types in Setting I on the CP-I cohort. Mental Health and Cognitive Specialty notes consistently demonstrated stronger predictive performance across timepoints.

### Stratified analysis by subgroups

To identify predictive features for specific populations, we performed stratified analyses by age, sex, and race/ethnicity. We trained models for different time points, focusing on the -1-day model for feature importance analysis due to the sparsity of subgroup populations. The conclusions below are statistically significant under DeLong test with p-value < 0.05. A minimal or no difference means p-value > =0.05 under the DeLong test.

### Stratification by age

Age is a well-established risk factor for AD[35], so we trained models on different age groups. For the -1-day prediction, performance was slightly lower in patients under 65 (AUROC 0.855) and over 85 (AUROC 0.851) compared to other age groups, though overall differences in performance and feature importance across groups were minimal.

### Stratification by sex

Recognizing sex differences in AD risk, we trained models separately for male and female groups. For females, the AUROC was 0.514 (-10 years) and 0.831 (-1 day), with AUPRC of 0.331 and 0.728. For males, the AUROC was 0.583 (-10 years) and 0.861 (-1 day), with AUPRC of 0.380 and 0.740.

Feature importance analysis of the -1-day model revealed sex-specific top predictors. For females, 'visuospatial ability' was a top ten predictor, but not for males. Conversely, 'mood' was a top ten predictor for males, but not for females. 'Fluency' was a top 20 predictor for females, while 'affect' was for males (Supplementary Fig. 17).

### Stratification by race/ethnicity

We stratified the cohort by race/ethnicity and trained separate models. For white patients, AUROC was 0.579 (-10 years) and 0.862 (-1 day), with AUPRC of 0.367 and 0.740. For Black/African American patients, AUROC was 0.564 (-10 years) and 0.867 (-1 day), with AUPRC of 0.343 and 0.763. 'Pain' was a top ten predictor for white patients, while 'getting lost' was prominent for Black/African American patients. In the top 20 predictors, 'affect' and 'memory' issues were significant for white patients, whereas 'delusions' and 'incontinence' were significant for Black/African American patients (Supplementary Fig. 18).

The model trained on Non-Hispanic/Latino patients achieved an AUROC of 0.570 (-10 years) and 0.865 (-1 day), with AUPRC of 0.365 and 0.742. For the Hispanic/Latino cohort, AUROC was 0.633 (-10 years) and 0.873 (-1 day), with AUPRC of 0.465 and 0.760. Analysis of feature importance of the -1-day model, 'communication' and 'pain' were among the top ten predictors for the Non-Hispanics/Latinos but not for the Hispanics/Latinos, while 'getting lost' and 'hearing' were among the top ten predictors for Hispanics/Latinos but not for Non-Hispanic/Latinos. Additionally, 'remembering', 'visuospatial ability', 'depression', and 'affect' were among the top 20 predictors unique to Non-Hispanic/Latino patients, whereas 'incontinence', 'wandering', 'delusion', and 'language' were among

the top 20 predictors unique to Hispanic/Latino patients (Supplementary Fig. 19).

## Discussion

With the aging population, the prevalence of AD is rapidly increasing[22]. Therapies to prevent or treat early AD appear increasingly important[42]. Effective use of these therapies, though, will depend on the efficient identification of high-risk individuals who might benefit best from therapies. Some efforts have identified predictive factors for AD progression[43] and some individual early identification tools have been developed[44], but to date, no efficient approach exists for large-scale screening of patients at high risk of AD.

Here, we explore whether routine information recorded in EHRs can help with this task. We found that keywords that may be signs or symptoms of AD dementia, or SCD, such as forgetfulness or depression, are routinely recorded in the EHRs of our studied patients. These keywords reflect a wide spectrum of physical and psychological conditions noted in routine patient health management.

Notably, around 15 years before an AD diagnosis was recorded in ICD codes, the frequency of relevant keywords in EHRs began to rise markedly compared to controls. This pattern suggests that patients were experiencing and reporting subjective symptoms—such as memory lapses, concentration difficulties, or mood changes—that clinicians deemed noteworthy enough to document. These reports often occurred well before objective cognitive impairment could be detected through formal assessments like the Mini-Mental State Examination (MMSE) or comprehensive neuropsychological batteries. While our EHR corpus included references to such assessments (e.g., MMSE, MoCA), we deliberately excluded them from the feature set. Our aim was to capture early, symptom-level indicators of AD dementia, as reflected in routine clinical documentation from patients, family members, or providers—prior to structured diagnostic evaluation. This approach is intended to detect subtle, early patterns that may reflect emerging cognitive or behavioral concerns not yet formally assessed.

In contrast to traditional methods that assess SCD through structured questionnaires administered during neuropsychological visits (e.g., asking, "Do you feel your memory is becoming worse?") our approach identifies similar subjective concerns embedded within routine clinical interactions. This offers a scalable, real-world alternative to detect early cognitive changes, potentially enabling earlier risk stratification and intervention in patients at risk of AD.

Importantly, we found that the occurrence, and increasing pattern, of these keywords is predictive of the development of AD. Our keyword-based approach significantly outperformed the structured data model for making predictions one day to ten years prior to diagnosis, highlighting the effectiveness of leveraging unstructured textual data from clinical notes. The performance was also validated using hold-out VHA stations and by age, sex, and race/ethnicity, demonstrating its robustness and generalizability. These findings suggest that using routine clinical care records to identify AD dementia-related signs and symptoms is a more effective method for predicting AD risk than relying on structured data of EHRs like diagnostic codes and medications etc. Keywords predictors appear to be more representative and indicative of AD dementia risk, as it allows for a wider net to capture SCD and AD dementia signs and symptoms that change in frequency.

Early identification of AD is crucial in primary care settings to facilitate timely referrals and treatments for high-risk patients[45]. By analyzing primary care notes, we found similar patterns of SCD and AD dementia signs and symptoms, aligning with existing work[46,47], showing that patients with early AD dementia signs usually first visit a primary care provider. Subtle changes in cognition or behavior may be noticed during routine wellness visits or appointments for other comorbidities. Although our model's performance was reduced when trained exclusively on primary care notes compared to a wider range of clinical notes, it still demonstrated meaningful predictive ability. This finding highlights the potential value of primary care documentation as a practical early screening resource. In real-world clinical

workflows, embedding automated AD risk detection models into primary care EHR systems could enable proactive identification of at-risk individuals without requiring specialized neurology or geriatrics consultations at the outset.

Future work could explore targeted screening strategies triggered by early narrative indicators in primary care notes, potentially facilitating earlier diagnostic evaluations, initiating preventive interventions, and improving long-term patient outcomes.

Our stratified analysis revealed that physiological and neuropsychiatric symptoms—such as sleep disturbances, mood changes, and behavioral alterations—tended to appear earlier in the clinical record than explicit memory complaints, which became more prominent closer to diagnosis. Additionally, impairments in activities of daily living (ADLs) emerged as strong predictive features, consistent with the progressive functional decline characteristic of AD. These findings align with and extend prior work[48] used machine learning to detect dementia from EHR and survey data, emphasizing the predictive value of non-cognitive features in early stages of the disease. Our results support the growing recognition that non-cognitive symptoms may serve as valuable early indicators of AD risk[49–51], particularly when structured testing is not yet initiated. Leveraging such symptoms from routine clinical narratives could enhance early detection strategies and complement traditional memory-focused assessments.

Early-onset AD (EOAD) patients—diagnosed before age 65—constituted 3.5% of our CP1 cohort. These patients had a higher average number of keyword mentions and a steeper increase in mentions as they approached diagnosis, consistent with literature suggesting EOAD patients experience a more rapid cognitive decline compared to late-onset AD patients[52].

Previous research indicates that women experience faster cognitive decline than men following a diagnosis of mild cognitive impairment (MCI) or AD dementia, and that males and females with AD exhibit different cognitive and psychiatric symptoms[53]. Our study investigates early AD dementia signs and symptoms trends prior to diagnosis, revealing that female AD patients have more frequent mentions of SCD and AD dementia keywords and a faster acceleration rate of these keywords before diagnosis. Feature importance analysis shows visuospatial ability is a predominant factor in female patients, while mood and affect are more significant in male patients.

When stratifying the patient cohort by race and ethnicity, we found that, in addition to Hispanic/Latino individuals, Black/African and other racial groups had more frequent mentions of AD-relevant keywords than non-Hispanic/Latino and White patients. Our predictive model performed better on the Hispanic/Latino population (AUROC of 0.633 at -10-year and 0.873 at -1-day) than on the entire cohort (AUROC of 0.577 at -10-year and 0.861 at -1-day). Feature importance analysis revealed disparities among the top predictors for Hispanic/Latino versus non-Hispanic/Latino patients, as well as between White and Black/African patients. Neuropsychiatric and cognition-related keywords were more prominent among the top predictors for non-Hispanic/Latino and White patients, while physiological change-related keywords appeared more frequently among the top predictors for Hispanic/Latino and Black patients. Studies estimated that the Latino community is 1.5 times more likely to develop AD than White non-Hispanics. Social determinants of health, comorbidities, and genetic factors contributed to the high incidence of AD in this group. Research also shows that Hispanic/Latino individuals may be more reluctant to seek medical help for cognitive impairment due to healthcare discrimination[54].

Our study has several limitations. First, we used VHA data, which may not represent the general population, as VHA patients tend to be demographically skewed, socioeconomically disadvantaged, and have a high prevalence of conditions, such as post-traumatic stress disorder and traumatic brain injury[55,56]. Our findings need validation in non-VHA populations for future work. Second, we conducted a case-control study, which is simpler than a cohort study. A cohort study was not feasible for this work due to the high computational cost of analyzing over 20 years of longitudinal EHR notes. Third, our study assumes that EHR notes accurately and

**Article**

comprehensively capture the historical symptoms of patients. However, EHRs are known to contain limitations, such as missing information and widespread use of copy-pasted content, especially in large healthcare systems where clinicians often reference prior documentation. These practices can obscure the true onset or context of symptoms. Additionally, we did not implement explicit strategies to exclude templated or auto-populated text, which may contain redundant or non-informative content that could impact symptom extraction. In addition, diagnosing AD is challenging. Although we implemented two CP rules for identifying AD cases, we acknowledge that some AD patients may receive delayed diagnoses or be misclassified with other types of dementia, leading to their exclusion from our study. Fourth, unlike previous studies excluding patients with less than 7 years of EHR history[18], we excluded those with less than 5 years. This may still introduce selection bias, as patients with lower hospital utilization and fewer clinical visits, who might benefit more from large-scale screening, were underrepresented in our study. We plan to explore additional data sources to include this group in our prediction model. Fifth, our analysis did not account for the effects of disease-modifying interventions. While effective disease-modifying therapies were not widely available during the majority of our study period, symptomatic treatments, such as donepezil were commonly prescribed and may have influenced the observed trends in SCD and AD dementia signs and symptoms, which are largely captured by our NLP-based extraction model. The availability and adoption of symptomatic interventions could alter the clinical presentation and progression patterns captured in EHR notes, potentially impacting the model's predictions. Additionally, our current NLP approach does not explicitly distinguish between symptom changes due to natural disease progression versus treatment effects. Future work could explore incorporating medication history and treatment response patterns into the modeling framework to better differentiate between disease-driven and therapy-modulated symptom trajectories. We will continue to monitor the impact of existing and emerging therapies on symptom documentation and model performance.

Sixth, our keyword-based approach for identifying SCD and AD-relevant symptoms does not incorporate advanced negation detection or contextual disambiguation methods. As a result, negated mentions or symptoms (e.g., "no memory loss," "denies confusion") documented in non-indicative contexts may be mistakenly extracted as positive signals. Although both case and control cohorts were processed identically—minimizing differential bias—this limitation may still affect the overall precision of symptom identification. Future work will explore the integration of negation detection algorithms and more sophisticated NLP techniques to improve accuracy.

Despite limitations, our study leverages a large national EHR dataset from the VHA, covering 1298 healthcare settings across the U.S. We utilized one of the largest patient cohorts for AD research, showing that SCD and signs and symptoms of AD dementia appear in clinicians' notes up to 15 years before diagnosis. NLP can detect these early signs, and keyword screening is more efficient than structured data and recent studies using predictors like grip strength, reaction time, prospective memory, and cognitive testing—measures not easily collected during routine visits[26]. Our machine learning approach aim to address gaps in testing accessibility and affordability, potentially enabling large-scale AD risk screening. Identifying predictive features across demographics offers insights for targeted prevention and awareness. These findings suggest that leveraging EHR notes could potentially be a powerful tool for early AD identification and management. In recent years, large language models (LLMs) have demonstrated strong performance across a variety of clinical applications, including clinical text understanding, information extraction, and decision support[57–60]. Given their ability to process complex narrative data, LLMs could play an important role in enhancing early AD risk detection by extracting subtle pre-diagnostic signals from unstructured EHR notes, operating effectively without requiring large annotated datasets, and generating human-interpretable rationales that support clinical trust. Looking ahead, future work could explore LLM-based multi-agent frameworks that simulate multidisciplinary diagnostic teams, offering a promising path to improve individualized risk assessment and early intervention strategies for AD.

## Data availability

The data used in this study are from the U.S. Veterans Health Administration. In accordance with federal regulations and Department of Veterans Affairs policies, these data cannot be publicly released. Interested investigators may request access to the underlying data, subject to VA approval; once such approval is granted, the corresponding author (Hong_Yu@uml.edu) can facilitate data sharing in compliance with VA requirements. No source data are applicable to Figs. 1 and 2 due to the nature of the flowchart and workflow. The source data for Figs. 3 and 4 are provided in Supplementary Data 1.

## Code availability

Code for prediction models can be made available upon request. Relevant packages include Python scikit-learn version 0.23.2, scipy version 1.2.0.

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

## Acknowledgements

This work was supported by the National Institutes of Health (R01AG080670, R01DA056470-A1) and the Department of Veterans Affairs Health Services Research program (I01HX003711-01A1, I02HX003969). The content is solely the responsibility of the authors and does not necessarily represent the official views of the NIH or the Department of Veterans Affairs. The funding agencies had no role in the design, conduct, data analysis, interpretation, or decision to publish this study.

## Author contributions

H.Y., D.B., and R.L. conceived and designed the study. R.L. drafted the manuscript. R.L. and X.W. contributed the experiments and performed data analyses and formatted results (figures, tables). W.H. and W.L. assisted with cohort creation and data preparation. W.H., H.K., and R.G. curated the keywords list. D.B., J.M., B.S., and H.L. offered critical comments and helpful suggestions, and critically revised the manuscript. J.M. and B.S. provided interpretation of study results and clinical subject matter expertise. H.Y. provided research guidance and supervision. R.L., X.W., and W.H. accessed and verified all underlying data for these analyses. All authors contributed to the editing of the manuscript. All authors agreed with the results and conclusions and approved the final draft. Authors had access to the data in the study, and they accept responsibility to submit for publication.

## Competing interests

The authors have no conflicts of interest.
