## [Peer Review file · Communications Medicine]

Early Prediction of Alzheimer's Disease using Longitudinal Electronic Health Records of US Military Veterans

Corresponding Author: Dr Hong YU

Version 0:

Reviewer comments:

Reviewer #1

(Remarks to the Author)

The manuscript titled " Prediction of Alzheimer's Disease (AD) Leveraging Symptom Occurrences from Longitudinal Electronic Health Records of US Military Veterans" presents a timely and relevant exploration of leveraging natural language processing (NLP) to detect cognitive decline using electronic health records (EHRs) clinical notes. This study presents an analysis of longitudinal EHRs from the Veterans Health Administration to predict AD onset using NLP on unstructured clinical notes. The study's core premise—that subjective cognitive decline (SCD) and AD-related symptoms appear in EHRs long before a formal diagnosis—has significant implications for and intervention. The authors demonstrate that keyword mentions increase exponentially in cases compared to controls, and that random forest (RF) models trained on these features outperform models using structured data alone.

While the manuscript offers valuable insights, there are several areas where additional clarity, adjustments, and consistency could enhance its overall quality and contribution to the field.

Introduction

- The introduction effectively contextualizes the study within the broader public health challenge of AD.
- The statement "The number of people with AD could rise from 6.9 million today to 13.8 million in the U.S. and over 100 million worldwide by 2060" lacks specificity due to the use of the word "today." It would be clearer to specify the exact year corresponding to the 6.9 million estimate.
- The phrase "potentially leading to new therapies that alter cognitive decline" is vague because the term "alter" is non-specific. It is unclear whether the intended meaning is to slow, halt, or reverse cognitive decline. Clarifying the specific impact of these therapies would improve the statement's precision and readability.

Methods

- The authors states that "Through extensive literature review, domain experts curated a list of 122 SCD and AD dementia-relevant keywords." However, the exact number of domain experts involved, and their areas of expertise are not specified. It is also unclear whether the list of keywords was generated through a formal consensus process (e.g., Delphi method) or if it was solely based on literature review. Including this information would enhance the transparency and reproducibility of the study.
- Clinical notes frequently contain templated phrases and structured documentation patterns, which can influence the accuracy of NLP-based symptom extraction. Many EHR systems use standardized templates that may pre-fill information, potentially leading to redundant or non-indicative mentions of symptoms. Additionally, negation phrases (e.g., "no evidence of memory impairment" or "denies cognitive difficulties") must be accurately detected to prevent false-positive symptom extractions. It is important to clarify how the NLP pipeline handled these challenges. Did the authors implement techniques such as template filtering, structured text exclusion, or rule-based methods to distinguish between manually entered notes and automated templates? Moreover, for negation, did they use established methods such as NegEx, Contextual Negation Models, or other syntactic approaches to ensure that negated symptoms were not incorrectly classified as present? Addressing these aspects would enhance confidence in the accuracy of extracted symptom mentions. If no explicit negation handling was performed, this should be acknowledged as a potential limitation, as it could impact the reliability of the extracted SCD and AD symptom data.

Results

- The study primarily relies on RF models but does not provide a justification for why this specific algorithm was chosen over

alternatives such as logistic regression, gradient boosting, or deep learning-based models. While RF models are useful for handling high-dimensional data, a comparative analysis of different classifiers would help contextualize its performance. The manuscript would benefit from an explicit discussion on why RF was deemed the most suitable approach.

- The manuscript states that “Existing work primarily utilized structured EHR data like medications, ICD codes, and abnormal measurements for AD prediction,²¹ We evaluated the random forest model using only these structured clinical features on our dataset.” However, the sentence structure is unclear and should be revised for better readability. Additionally, the results should specify whether models trained on structured data were evaluated separately or in combination with unstructured text features. If structured and unstructured features were tested independently, were there specific reasons for this approach? Given that a core premise of the study is the predictive value of unstructured text, it would be valuable to explain why the structured data model was not integrated with text-based features for a more comprehensive evaluation.
- Throughout the manuscript, the cohort names are inconsistently formatted as “CP-I” and “CP-II” in some instances and “CP1” and “CP2” in others. Consistency in notation should be maintained to avoid confusion.
- The manuscript reports prediction results at two specific time points: 10 years before diagnosis and 1 day before diagnosis. However, it is unclear why these particular intervals were chosen. Were these time points selected based on clinical relevance, data availability, or computational constraints? Providing a rationale for these choices would improve the interpretability of the results. Additionally, it would be useful to explore intermediate time points (e.g., 5 years before diagnosis) to assess the progression of predictive performance over time.
- In the stratification analysis, the manuscript reports AUROC values and mentions statistical significance in some cases but does not provide details on how significance was measured. Were statistical tests (e.g., DeLong’s test for AUROC comparison) performed to assess differences? If so, were they applied uniformly across all stratifications? A more transparent discussion of significance testing is necessary for rigorous evaluation.

Discussion

- The analysis on primary care notes provides an insightful observation. Consider further discussing how this finding could support targeted screening in primary care settings.
- The authors acknowledge that “Our analysis did not account for effective disease-modifying interventions, but symptomatic treatments like donepezil were widely used and may have influenced observed trends in SCD and AD dementia signs and symptoms.” This is an important consideration, and further discussion on whether NLP models could distinguish between disease progression and treatment effects would be beneficial.
- The manuscript discusses the promise of machine learning in large-scale AD risk screening but does not mention the potential role of large language models (LLMs). Given recent advancements in transformer-based models for clinical text analysis (e.g., BioBERT, GPT-based approaches), discussing how LLMs could refine or enhance predictions would add a forward-looking perspective to the study.

Reviewer #2

(Remarks to the Author)

The principle claim of this paper is that the occurrence of certain key words in electronic patient records may contribute to the accuracy of decision support mechanisms aimed at identifying patients with degenerative disorders such as Alzheimer’s disease at earlier stages of their disease course. If they are accurate, ‘warning systems’ that alert a clinician to the possible presence of a disease state or prognostic indicator could improve safety, efficacy, efficiency and equity in clinical practice. If unreliable or inaccurate, however, they could have negative effects on all these measures.

It therefore seems reasonable to report the relative rates of appearance over time of key words describing cognitive symptoms in the electronic health records of a large and relatively homogeneous patient population, to group these into categories that reflect different aspects of dementia patients’ experience, and to measure the accuracy with which they distinguish AD patients from matched controls at different time points before the diagnosis of AD is made. Using strict case identification criteria and an appropriate and rigorously tested classification algorithm, keyword features achieve an acceptable level of discrimination in the data used, up to 3 years before the diagnosis is made.

The authors provide a long and comprehensive list of caveats to the interpretation of their results, including the possibility that the model may not be generalisable to other datasets, all of which are valid, even if not (yet) fatal criticisms of the approach. In addition to these, though, I would add the following observations:

- 1) Why and on what basis did the authors choose these specific keywords? The only clue to this critical stage in the methodology is the input of ‘extensive literature review [by] domain experts’, though it was hard to see how the short list of publications on which this review was based would help anyone, however expert, to identify the words most likely to be used to describe cognitive symptoms by undiagnosed individuals with incipient Alzheimer’s disease. A more valid (if more time-consuming) approach would surely have been to calculate keyness values for all words that appeared in the AD patients’ records using the control records as a reference corpus, and to set a threshold value and ask the domain experts to identify terms that were relevant to cognitive symptomatology.
- 2) Even if the bottom-up approach recommended above is not considered appropriate to the question or feasible in the context, the authors should consider conducting similar analyses using ‘control’ keywords, relevant, for example to conditions that are independent of dementia - such as joint pain or skin complaints. Absence of a difference between the groups in the rate of appearance and change over time would make the idea that the selected keywords have specific value.
- 3) The authors recognise the importance of normalising the rates of keyword appearance and report the change in these in one, but not all, of their results. It is not clear whether the critical AUROC and AUPRC statistics reported in Tables 2 (a-c) are derived from these normalised rates or from absolute counts. The longitudinal plots in Figures 3 (a-h) represent changes in

absolute counts rather than normalised rates.

Reviewer #3

(Remarks to the Author)

This manuscript describes findings from an observational study designed to use machine learning (ML) approaches to detect Alzheimer disease (AD) diagnosis many years prior. The authors used a case-control design using data from the VA EHR database. They identified all AD diagnostic cases from VA EHR and then matched 4:1 with Veterans without non-dementia diagnoses based on demographics and clinical utilization, ADI and comorbidity index. The goal was to use ML approaches to detect signs/symptoms from EHR prior to diagnosis of AD. They used natural language processing (NLP) from a list of 122 key words (grouped by category). They found that cases with AD had greater increase in relevant words compared to controls and that the AD group increased exponentially prior to diagnosis. These patterns were fairly similar by VA site and for strata of age, sex and race/ethnicity.

There is no "methods" section of the manuscript.

Subjective cognitive decline (SCD) is a bit controversial as a precursor to AD as fairly nonspecific and often associated with psychiatric and other medical conditions. The connection between mild cognitive impairment (MCI) is much stronger than for SCD. It is surprising they did not use MCI instead. The goal of using SCD was not clear really and distracting and not sure how it helps here.

Things really differed (exponentially) around 2 years prior to diagnosis (figure 3). This is interesting and it might make sense to look more closely at this time point.

Along those lines, it is unclear what -1 day means and why choose 10 years before and then 1 day before diagnosis if that is what it means. The results are strikingly different. 1 day prior to diagnosis is unlikely to be that helpful for prediction and the AUC is much higher. I would like to see the results at 2 years prior as that seems to be a real inflection point but that was not investigated (see Figure 4).

The 10-year data is of interest but pretty low AUC. I think emphasizing the few years prior to diagnosis is more important. The 1-day prior to diagnosis had much higher AUC but then much less helpful prognostically.

Interestingly, the physiological changes and neuropsychiatric symptoms are the most noted and then memory changes closer to diagnosis. ADLs are also big driver. It is important to reference some prior work by L Cleret de Langavant who used ML to identify dementia cases from EHR and other survey data. This idea of non-cognitive symptoms being useful on is a key message and others have reported on this.

Please elaborate on differences by clinic/specialty. There is one figure referring to primary care only.

Figure 3 is a bit redundant...some can be put in supplement or mentioned in text. There are way too many figures anyway.

Version 1:

Reviewer comments:

Reviewer #1

(Remarks to the Author)

The authors addressed my comments well. I recommend acceptance.

Reviewer #4

(Remarks to the Author)

I have gone through the comments of all previous reviewers and the author responses to those comments. All my comments have been covered by previous reviewers and I am fully satisfied with the author responses. I have nothing further to add.

Reviewer #5

(Remarks to the Author)

I have carefully reviewed your responses to Reviewer #3's comments. I find that you have addressed the points raised in a thorough and convincing manner. The clarifications and additional analyses provided are appropriate and satisfactory, and I do not have further concerns.

Reviewers'

comments:

Reviewer #1 (Remarks to the Author):

The manuscript titled " Early Prediction of Alzheimer’s Disease (AD) Leveraging Symptom Occurrences from Longitudinal Electronic Health Records of US Military Veterans" presents a timely and relevant exploration of leveraging natural language processing (NLP) to detect cognitive decline using electronic health records (EHRs) clinical notes. This study presents an analysis of longitudinal EHRs from the Veterans Health Administration to predict AD onset using NLP on unstructured clinical notes. The study's core premise—that subjective cognitive decline (SCD) and AD-related symptoms appear in EHRs long before a formal diagnosis—has significant implications for early detection and intervention. The authors demonstrate that keyword mentions increase exponentially in cases compared to controls, and that random forest (RF) models trained on these features outperform models using structured data alone.

While the manuscript offers valuable insights, there are several areas where additional clarity, adjustments, and consistency could enhance its overall quality and contribution to the field.

Introduction

- The introduction effectively contextualizes the study within the broader public health challenge of AD.

Response:

Thank you for your positive feedback. We’re glad to hear that the introduction clearly conveyed the public health significance of AD and contextualized our study accordingly.

- The statement “The number of people with AD could rise from 6.9 million today to 13.8 million in the U.S. and over 100 million worldwide by 2060” lacks specificity due to the use of the word "today." It would be clearer to specify the exact year corresponding to the 6.9 million estimate.

Response:

Thank you for this helpful suggestion. The estimate of 6.9 million people living with AD in the U.S. is based on the *2024 Alzheimer’s Disease Facts and Figures* report, which we cited. We agree that specifying the year improves clarity and have revised the sentence accordingly by replacing “today” with the exact year:

Revised sentence:

“In 2024, an estimated 6.9 million people in the U.S. are living with AD, and this number is projected to rise to 13.8 million in the U.S. and over 100 million worldwide by 2060.”²

- The phrase “potentially leading to new therapies that alter cognitive decline” is vague because the term “alter” is

non-specific. It is unclear whether the intended meaning is to slow, halt, or reverse cognitive decline. Clarifying the specific impact of these therapies would improve the statement's precision and readability.

Response:

Thank you for your insightful comment. We agree that the term “alter” was too vague and could lead to ambiguity. To enhance clarity and precision, we have revised the sentence to more clearly reflect the intended therapeutic impact.

Revised sentence:

Early prediction of AD risk is crucial for effective interventions and treatments, potentially enabling the development of therapies that slow or halt cognitive decline.¹⁻⁴

Methods

- The authors states that “Through extensive literature review, domain experts curated a list of 122 SCD and AD dementia-relevant keywords.” However, the exact number of domain experts involved, and their areas of expertise are not specified. It is also unclear whether the list of keywords was generated through a formal consensus process (e.g., Delphi method) or if it was solely based on literature review. Including this information would enhance the transparency and reproducibility of the study.

Response:

Thank you for your thoughtful comment. We agree that additional detail is needed to enhance the transparency and reproducibility of our keyword development process. In response, we have clarified the number of domain experts involved, their areas of expertise, and the approach used to generate and refine the final list.

Specifically, the list of 122 SCD- and AD dementia–relevant keywords was developed with input from six domain experts: three health specialists with expertise in cognitive aging and EHR-based research, two neurologists, and one epidemiologist. Wang et al. (2021) demonstrated that incorporating expert-curated keyword lists improved the identification of cognitive decline cases from unstructured clinical note sections (JAMA Network Open, 4(11): e2135174). Inspired by this precedent, each health specialist independently proposed candidate keywords based on their experience reviewing clinical narratives and a targeted literature review. The neurologists and epidemiologist provided clinical oversight and contributed through weekly discussions, ensuring the clinical accuracy, consistency, and relevance of the proposed terms. The final list was consolidated through collaborative discussion and mutual agreement among all experts. While structured and iterative, this process did not follow a formal Delphi method.

The manuscript has been updated accordingly to reflect these clarifications and supporting evidence.

Revision in manuscript:

Building on an extensive literature review, and inspired by the expert-curated keyword approach demonstrated by Wang et al. (2021), we curated a list of 122 SCD- and AD dementia–relevant keywords through a structured, expert-driven process. Six domain experts contributed to the development of this list: three health specialists with expertise

in cognitive aging and EHR-based research, two neurologists, and one epidemiologist. The health specialists independently proposed candidate terms based on their clinical experience reviewing narrative EHRs and insights from a targeted literature review. The neurologists and epidemiologist provided clinical oversight through weekly meetings, helping validate the accuracy, consistency, and clinical relevance of the terms. The final list was refined through collaborative discussion and mutual agreement. Keywords were then grouped into clinically meaningful categories, including cognition–speech/language, cognition–memory, cognition–learning/perception, assistance needed, physiological changes, and neuropsychiatric symptoms (Box 1).

- Clinical notes frequently contain templated phrases and structured documentation patterns, which can influence the accuracy of NLP-based symptom extraction. Many EHR systems use standardized templates that may pre-fill information, potentially leading to redundant or non-indicative mentions of symptoms. Additionally, negation phrases (e.g., “no evidence of memory impairment” or “denies cognitive difficulties”) must be accurately detected to prevent false-positive symptom extractions. It is important to clarify how the NLP pipeline handled these challenges. Did the authors implement techniques such as template filtering, structured text exclusion, or rule-based methods to distinguish between manually entered notes and automated templates? Moreover, for negation detection, did they use established methods such as NegEx, Contextual Negation Models, or other syntactic approaches to ensure that negated symptoms were not incorrectly classified as present? Addressing these aspects would enhance confidence in the accuracy of extracted symptom mentions. If no explicit negation handling was performed, this should be acknowledged as a potential limitation, as it could impact the reliability of the extracted SCD and AD symptom data.

Response:

Thank you for your thoughtful and insightful comments. We fully agree that accurately handling templated text and negation in clinical notes is critical, as both can significantly impact the precision of NLP-based symptom extraction. While our original submission briefly acknowledged related limitations—namely, (1) the assumption that EHRs reliably capture symptom history despite missing information and copy-pasted content, and (2) the inherent imprecision of relying solely on keyword matching—we appreciate the reviewer’s detailed articulation of these challenges.

In the current version of our pipeline, we did not implement explicit techniques such as template filtering, structured text exclusion, or rule-based negation detection (e.g., NegEx or syntactic methods). Our primary aim was to design a simple and computationally efficient approach that approximates symptom presence using keyword matching. Notably, both AD case and control cohorts were processed identically, so any inaccuracies introduced by templated language or negated phrases would be expected to impact both groups similarly. As a result, while some individual symptom mentions may be misclassified, we believe the group-level comparisons and overall conclusions remain valid.

That said, we fully acknowledge that incorporating more rigorous symptom extraction strategies would improve the reliability and granularity of our findings. We have updated the manuscript’s limitations section to explicitly reflect this and identified it as a key direction for future work. In particular, we plan to explore the integration of template

filtering heuristics, section-aware parsing, and rule-based or transformer-based negation detection methods to better distinguish between meaningful clinical content and non-informative or negated text.

We appreciate the reviewer's thoughtful feedback, which has helped us strengthen both the transparency and future direction of our methodological approach.

Revision in manuscript:

Third, our study assumes that EHR notes accurately and comprehensively capture patients' historical symptoms. However, EHRs often have limitations, such as missing information and the widespread use of copy-pasted content—especially in large healthcare systems where clinicians frequently reference prior documentation. These practices can obscure the true onset or context of symptoms. We also did not implement explicit strategies to exclude templated or auto-populated text, which may include redundant or non-informative content that could affect symptom extraction.

Sixth, our keyword-based approach for identifying SCD and AD-relevant symptoms does not incorporate advanced negation detection or contextual disambiguation methods. As a result, negated mentions or symptoms (e.g., “no memory loss,” “denies confusion”) documented in non-indicative contexts may be mistakenly extracted as positive signals. Although both case and control cohorts were processed identically—minimizing differential bias—this limitation may still affect the overall precision of symptom identification. Future work will explore the integration of negation detection algorithms and more sophisticated NLP techniques to improve accuracy.

Results

- The study primarily relies on RF models but does not provide a justification for why this specific algorithm was chosen over alternatives such as logistic regression, gradient boosting, or deep learning-based models. While RF models are useful for handling high-dimensional data, a comparative analysis of different classifiers would help contextualize its performance. The manuscript would benefit from an explicit discussion on why RF was deemed the most suitable approach.

Response:

Thank you for these thoughtful and constructive comments. We fully agree that clearer justification of our model choice is important, and we appreciate the opportunity to clarify this in the revised manuscript.

As noted in the manuscript, existing work such as Tang et al. (2024) also adopts RF for AD prediction. RF offers a strong balance between interpretability and predictive flexibility—providing greater transparency than neural networks while capturing nonlinear relationships more effectively than logistic regression. This interpretability is especially valuable in clinical contexts, where understanding feature contributions is as important as achieving high predictive accuracy.

In addition to RF, we added logistic regression and XGBoost performance as baseline models (Supplementary Table 4). RF consistently outperformed logistic regression, while XGBoost yielded comparable performance and was less interpretable—particularly when applied to sparse, TF-IDF-based keyword features. Given our goal of developing a

simple and computationally efficient approach, we did not pursue deep learning–based classifiers, as they are computationally intensive, require large, dense datasets, and offer limited interpretability—making them less suitable for our sparse feature set and relatively moderate training sample size. Traditional statistical models better align with the goals and constraints of this work.

We have revised the manuscript to explicitly discuss these considerations and better contextualize our choice of RF. Thank you again for these important suggestions, which have helped strengthen the methodological clarity of the study.

Supplementary Table 4: Logistic regression and XGBoost prediction results using keyword features in Setting I on the CP-I cohort

Prediction Year	Logistic Regression		XGBoost	
	AUROC	AUPRC	AUROC	AUPRC
-1 day	0.826	0.670	0.861	0.741
-1 year	0.745	0.545	0.772	0.603
-2 year	0.707	0.483	0.731	0.532
-3 year	0.679	0.438	0.705	0.488
-5 year	0.636	0.369	0.663	0.417
-7 year	0.607	0.331	0.628	0.369
-10 year	0.570	0.356	0.577	0.373

• The manuscript states that “Existing work primarily utilized structured EHR data like medications, ICD codes, and abnormal measurements for AD prediction,²¹ We evaluated the random forest model using only these structured clinical features on our dataset.” However, the sentence structure is unclear and should be revised for better readability. Additionally, the results should specify whether models trained on structured data were evaluated separately or in combination with unstructured text features. If structured and unstructured features were tested independently, were there specific reasons for this approach? Given that a core premise of the study is the predictive value of unstructured text, it would be valuable to explain why the structured data model was not integrated with text-based features for a more comprehensive evaluation.

Response:

Thank you for this helpful observation. We agree that the original sentence could be improved for clarity. We have revised it in the manuscript for better readability.

We appreciate the opportunity to clarify the evaluation framework. In the study, we separately evaluated models trained on: (1) structured clinical features only (medications, ICD codes, abnormal lab measurements); (2) keyword-derived features from unstructured clinical notes only; and (3) the combination of both structured and keyword-based features. These results were presented in Tables 2(a), 2(b), and 2(c), respectively.

Our rationale was to isolate and compare the predictive value of structured data alone, unstructured text features alone, and the integrated feature set. This design allowed us to directly assess the added value of mining narrative notes relative to traditional structured fields, which is a core focus of the study. We have clarified this point in the revised manuscript.

Thank you again for this important suggestion, which has helped us improve the clarity and completeness of the methodology description.

Revision in manuscript:

Existing work primarily utilized structured EHR data like medications, ICD codes, and abnormal lab measurements for AD prediction,²¹ we evaluated the predictive performance of a random forest model using three distinct feature sets: (1) structured clinical features only (i.e., medications, ICD codes, and abnormal lab measurements), (2) keyword-derived features extracted from unstructured clinical notes, and (3) a combination of structured and keyword-based features.

Using only structured EHR features, the model achieved an AUROC of 0.497 at -10 years and 0.682 at -1 day, with corresponding AUPRCs of 0.371 and 0.571 on the CP-I cohort (Table 2(a)). When using only keyword-derived features, the model substantially outperformed the structured-data-only approach, achieving an AUROC of 0.577 at -10 years and 0.861 at -1 day, and AUPRCs of 0.373 and 0.741 on the CP-I cohort (Table 2(b)). Similar improvements were observed in the CP-II cohort, where AUROCs reached 0.598 at -10 years and 0.896 at -1 day. Combining structured features with keyword-derived features provided modest additional gains, with AUROCs increasing to 0.581 at -10 years and 0.867 at -1 day, and AUPRCs to 0.376 and 0.747 on the CP-I cohort (Table 2(c)).

These results highlight that narrative symptom evidence extracted from unstructured clinical notes offers significant predictive value for early AD detection, even when used independently of traditional structured EHR fields. In addition to reporting results at the boundary time points of 10 years and 1 day before diagnosis, we also evaluated model performance at intermediate intervals of -1 year, -2 years, -3 years, -5 years, and -7 years prior to diagnosis, as detailed in Table 2(a-c).

- Throughout the manuscript, the cohort names are inconsistently formatted as “CP-I” and “CP-II” in some instances and “CP1” and “CP2” in others. Consistency in notation should be maintained to avoid confusion.

Response:

Thank you for pointing out this inconsistency. We agree that consistent formatting of cohort names is important for clarity. In the revised manuscript, we have standardized the naming convention throughout to consistently use “CP-I” and “CP-II” when referring to the two cohorts. We appreciate the reviewer’s careful reading, which has helped improve the precision and readability of the manuscript.

- The manuscript reports prediction results at two specific time points: 10 years before diagnosis and 1 day before diagnosis. However, it is unclear why these particular intervals were chosen. Were these time points selected based on clinical relevance, data availability, or computational constraints? Providing a rationale for these choices would improve the interpretability of the results. Additionally, it would be useful to explore intermediate time points (e.g., 5 years before diagnosis) to assess the progression of predictive performance over time.

Response:

Thank you for this thoughtful comment. We appreciate the opportunity to clarify our rationale for the choice of time points and prediction windows.

In the manuscript, we highlighted the 10-year and 1-day prediction results because these represent the two boundaries of our prediction horizon—the longest and shortest time intervals before formal diagnosis. This framing also reflects the natural performance boundaries of the models. Specifically:

- **10 years before diagnosis** represents the longest feasible prediction span given our dataset. For this task, only patient EHR notes up to 10 years prior to the first recorded AD diagnosis were used. To ensure sufficient information for feature extraction (requiring at least 5 years of clinical history), patients would need at least 15 years of longitudinal EHR data before their first AD diagnosis. Due to data availability constraints, very few patients have longer histories (e.g., ≥ 16 –20 years), making prediction at 11 years, 12 years, or beyond impractical.

- **1 day before diagnosis** represents the upper limit of our prediction task. Predictions made within 1 day of diagnosis are no longer considered “early” in a clinical sense, so -1 day serves as a natural endpoint for evaluating near-diagnostic performance.

Previous work by Tang et al. (2024) also reported similar time points in their manuscript (-1-day, -7-year).

We agree with the reviewer’s suggestion to explore intermediate time points to better characterize performance progression. As shown in Table 2(a,b,c,d), we reported model results at additional intermediate points: -1 year, -2 years (added in the revision), -3 years, -5 years, and -7 years before diagnosis. This provides a comprehensive view of how model performance changes as the time-to-diagnosis narrows.

We have revised the manuscript to clarify our rationale for time point selection and explicitly mention the intermediate time points reported in the tables. Thank you for this helpful suggestion, which improved the clarity of our study design.

- In the stratification analysis, the manuscript reports AUROC values and mentions statistical significance in some cases but does not provide details on how significance was measured. Were statistical tests (e.g., DeLong’s test for AUROC comparison) performed to assess differences? If so, were they applied uniformly across all stratifications? A more transparent discussion of significance testing is necessary for rigorous evaluation.

Response:

To evaluate differences in model performance across stratified subgroups, we used DeLong’s test for comparing ROC curves. In our stratification analysis (e.g., by age group, sex, and clinical specialty), we applied DeLong’s test pairwise to compare AUROC values between subgroup vs overall (male vs female for gender). All tests were two-sided, and p-values < 0.05 were considered statistically significant. The `roc_test` function from the `pROC` package in R was used to perform the comparisons.

Discussion

- The analysis on primary care notes provides an insightful observation. Consider further discussing how this finding could support targeted screening in primary care settings.

Response:

Thank you for this insightful suggestion. We agree that the predictive value observed from primary care notes has important implications for real-world screening. In the revised manuscript, we have expanded the Discussion to highlight how primary care documentation could serve as an effective foundation for early AD risk detection. Although model performance was lower when restricted to primary care notes compared to the full range of clinical specialties, it remained meaningful. This suggests that early cognitive or behavioral signs—often first recorded during routine wellness visits—can be leveraged for targeted screening strategies within primary care EHR systems. Such approaches could facilitate earlier identification, referrals, and interventions for at-risk individuals. We appreciate the reviewer’s comment, which helped us strengthen the clinical relevance and future application discussion.

- The authors acknowledge that “Our analysis did not account for effective disease-modifying interventions, but symptomatic treatments like donepezil were widely used and may have influenced observed trends in SCD and AD dementia signs and symptoms.” This is an important consideration, and further discussion on whether NLP models could distinguish between disease progression and treatment effects would be beneficial.

Response:

Thank you for this important observation. We fully agree that the presence of symptomatic treatments could impact the clinical manifestation of SCD and AD dementia signs and symptoms captured through NLP. In the revised manuscript, we have expanded the Limitations section to explicitly address this point. Specifically, we note that while our model captures symptom trajectories from clinical notes, it does not currently distinguish between changes driven

by natural disease progression versus those influenced by therapeutic interventions. Furthermore, we discuss how the availability and adoption of symptomatic therapies such as donepezil during the study period may have affected symptom patterns and, consequently, model predictions. We also outline that future work could incorporate medication history and treatment response modeling to better separate treatment effects from disease progression. We appreciate the reviewer's thoughtful suggestion, which helped us strengthen the discussion around this important limitation.

- The manuscript discusses the promise of machine learning in large-scale AD risk screening but does not mention the potential role of large language models (LLMs). Given recent advancements in transformer-based models for clinical text analysis (e.g., BioBERT, GPT-based approaches), discussing how LLMs could refine or enhance predictions would add a forward-looking perspective to the study.

Response:

Thank you for this thoughtful suggestion. We agree that large language models (LLMs) have strong potential to further advance AD risk prediction, particularly in extracting early signals from unstructured EHR narratives. In the revised manuscript, we have expanded the Discussion to include a forward-looking perspective on the role of LLMs. Specifically, we note that LLMs offer key advantages, including: (1) the ability to extract subtle, pre-diagnostic clinical indicators (e.g., memory lapses, mood changes) that are often overlooked in structured data; (2) effectiveness in low-resource settings, performing well even under zero-shot or few-shot conditions; and (3) the generation of human-interpretable rationales alongside binary predictions, which enhances clinical trust in AI-assisted decision-making.

Additionally, we would like to note that in a separate, ongoing project, we are developing a multi-agent LLM-based framework that simulates a multidisciplinary diagnostic process. Within this framework, specialized LLM agents—each focused on domains such as primary care, geriatrics, psychiatry, neurology, and clinical psychology—collaborate to conduct comprehensive longitudinal analyses of patient narratives. Early results from this work demonstrate the promise of LLM-driven, domain-specialized collaboration in improving individualized AD risk assessment. While beyond the scope of the current manuscript, we believe this line of research aligns closely with the reviewer's vision for the future application of LLMs in large-scale AD risk screening.

We appreciate the reviewer's insightful comment, which has strengthened the manuscript's forward-looking discussion.

Revision in manuscript:

In recent years, large language models (LLMs) have demonstrated strong performance across a variety of clinical applications, including clinical text understanding, information extraction, and decision support.⁴⁵⁻⁴⁸ Given their ability to process complex narrative data, LLMs could play an important role in enhancing early AD risk detection by extracting subtle pre-diagnostic signals from unstructured EHR notes, operating effectively without requiring large annotated datasets, and generating human-interpretable rationales that support clinical trust. Looking ahead, future

work could explore LLM-based multi-agent frameworks that simulate multidisciplinary diagnostic teams, offering a promising path to improve individualized risk assessment and early intervention strategies for AD.

Reviewer #2 (Remarks to the Author):

The principle claim of this paper is that the occurrence of certain key words in electronic patient records may contribute to the accuracy of decision support mechanisms aimed at identifying patients with degenerative disorders such as Alzheimer's disease at earlier stages of their disease course. If they are accurate, 'early warning systems' that alert a clinician to the possible presence of a disease state or prognostic indicator could improve safety, efficacy, efficiency and equity in clinical practice. If unreliable or inaccurate, however, they could have negative effects on all these measures.

It therefore seems reasonable to report the relative rates of appearance over time of key words describing cognitive symptoms in the electronic health records of a large and relatively homogeneous patient population, to group these into categories that reflect different aspects of dementia patients' experience, and to measure the accuracy with which they distinguish AD patients from matched controls at different time points before the diagnosis of AD is made. Using strict case identification criteria and an appropriate and rigorously tested classification algorithm, keyword features achieve an acceptable level of discrimination in the data used, up to 3 years before the diagnosis is made. The authors provide a long and comprehensive list of caveats to the interpretation of their results, including the possibility that the model may not be generalisable to other datasets, all of which are valid, even if not (yet) fatal criticisms of the approach. In addition to these, though, I would add the following observations:

1) Why and on what basis did the authors choose these specific keywords? The only clue to this critical stage in the methodology is the input of 'extensive literature review [by] domain experts', though it was hard to see how the short list of publications on which this review was based would help anyone, however expert, to identify the words most likely to be used to describe cognitive symptoms by undiagnosed individuals with incipient Alzheimer's disease. A more valid (if more time-consuming) approach would surely have been to calculate keyness values for all words that appeared in the AD patients' records using the control records as a reference corpus, and to set a threshold value and ask the domain experts to identify terms that were relevant to cognitive symptomatology.

Response:

Thank you for this thoughtful and important comment. We fully agree that the development of the keyword list is a critical component of our methodology, and we appreciate the opportunity to clarify this process to enhance its transparency and reproducibility.

Specifically, the list of 122 SCD- and AD dementia-relevant keywords was developed with input from six domain experts: three health specialists with expertise in cognitive aging and EHR-based research, two neurologists, and one epidemiologist. Wang et al. (2021) demonstrated that incorporating expert-curated keyword lists improved the

identification of cognitive decline cases from unstructured clinical note sections (JAMA Network Open, 4(11): e2135174). Inspired by this precedent, each health specialist independently proposed candidate keywords based on their experience reviewing clinical narratives and a targeted literature review. The neurologists and epidemiologist provided clinical oversight and contributed through weekly discussions, ensuring the clinical accuracy, consistency, and relevance of the proposed terms. The final list was consolidated through collaborative discussion and mutual agreement among all experts.

We also appreciate the reviewer's suggestion to consider a fully data-driven keyness analysis. However, applying such an approach in our context could have introduced circularity—specifically, the risk of bias by extracting keywords directly from the same dataset used for evaluation. To maintain methodological rigor, we intentionally separated the keyword curation, thus avoiding potential overfitting. By grounding our keyword selection in external clinical expertise and literature, we aimed to generate an independent and clinically meaningful feature set suitable for systematic validation.

We have revised the Methods section to include these clarifications. We appreciate the reviewer's insightful feedback, which has helped improve the clarity and rigor of our methodological description.

2) Even if the bottom-up approach recommended above is not considered appropriate to the question or feasible in the context, the authors should consider conducting similar analyses using 'control' keywords, relevant, for example to conditions that are independent of dementia - such as joint pain or skin complaints. Absence of a difference between the groups in the rate of appearance and change over time would make the idea that the selected keywords have specific value.

Response:

Thank you for this thoughtful and constructive suggestion. We agree that analyzing “control” keywords unrelated to AD dementia is a valuable way to assess the specificity of our symptom-based findings. While we initially considered joint pain, a literature review revealed consistent evidence of its positive association with dementia risk (e.g., Ikram et al., 2019; Kim et al., 2018), suggesting it may not serve as a truly independent control.

Instead, we conducted an analysis using a curated set of skin complaint/ Dermatologic-related control keywords—terms that are commonly documented in clinical practice but not known to be associated with cognitive decline, following your suggestion. These include general symptoms (e.g., *itching, rash*), localized findings (e.g., *skin lesion, ulcer*), and dermatologic diagnoses (e.g., *eczema, cellulitis*), selected through a targeted review of clinical sources and screened for clinical relevance, frequency in EHRs, and independence from AD dementia-related pathology. Our added keywords are shown in Supplementary Box 1.

As shown in the newly added Supplementary Figure 4, which presents the average number of dermatologic-related keywords normalized by notes No. by year before diagnosis, the longitudinal trends between AD cases and controls were largely similar, with no notable divergence over time. This supports the idea that the temporal patterns observed in our curated AD dementia-related keywords are not attributable to general shifts in health documentation or nonspecific symptom reporting.

We have added the results of this control analysis to the revised manuscript (see *Results, Supplementary Note 1, Supplementary Figure 4*) and appreciate the reviewer's insightful recommendation, which has strengthened the specificity and interpretability of our findings.

Dermatologic-related keywords:

General Symptoms: rash, itching/pruritus, dry skin, flaky skin, redness, skin irritation, cracked skin, skin pain, bruising, swelling, tender skin, burning sensation, oozing skin, inflamed skin, blister, peeling skin

Localized Findings: skin lesion, skin ulcer/ulcer, skin lump/bump, sore, scab, abscess, skin tag, mole, wart

Diagnoses / Conditions: eczema, psoriasis, acne, hives, dermatitis, contact dermatitis, atopic dermatitis, cellulitis, fungal infection, ringworm (tinea), scabies, shingles (herpes zoster), folliculitis, skin infection

Supplementary Figure 4. Dermatologic-related keyword patterns in longitudinal EHRs

Plots of dermatologic-related keyword counts for AD cases and matched controls: Average keywords per note by year before diagnosis.

3) The authors recognise the importance of normalising the rates of keyword appearance and report the change in these in one, but not all, of their results. It is not clear whether the critical AUROC and AUPRC statistics reported in Tables 2 (a-c) are derived from these normalised rates or from absolute counts. The longitudinal plots in Figures 3 (a-h) represent changes in absolute counts rather than normalised rates.

Response

Thank you for this important comment. We clarify that all longitudinal plots in Figure 4 (a-c) and Supplementary

Figure 1(a-e) (original Figures 3(a-h)) are normalized, though by different denominators based on analytical objectives. Figures 4(a), 4(b), and Supplementary Figure 1(a-e) present average AD dementia-related keyword counts per patient per year, which accounts for cohort size over time and ensures stability across demographic subgroups. We avoided normalizing by note count in these figures due to data sparsity—particularly in minor demographic groups—which could introduce noise.

To illustrate documentation intensity and symptom trends relative to clinical activity, Figure 4(c) presents keyword counts per note per year, revealing a similar increasing pattern for AD cases compared to controls. This was included intentionally to show that increasing keyword frequency is not solely due to increased patient visits.

Regarding the AUROC and AUPRC values in Tables 2(a-c), these were not derived from raw keyword counts. Instead, we used term frequency–inverse document frequency (TF-IDF) to represent AD dementia-relevant keywords extracted from EHR notes prior to diagnosis. This approach considers both the frequency of symptom mentions and their discriminative value, allowing our random forest model to capture meaningful, longitudinal patterns while down-weighting non-specific terms. We have updated the Methods and figure captions to clarify these points.

Reviewer #3 (Remarks to the Author):

This manuscript describes findings from an observational study designed to use machine learning (ML) approaches to detect Alzheimer disease (AD) diagnosis many years prior. The authors used a case-control design using data from the VA EHR database. They identified all AD diagnostic cases from VA EHR and then matched 4:1 with Veterans without non-dementia diagnoses based on demographics and clinical utilization, ADI and comorbidity index. The goal was to use ML approaches to detect early signs/symptoms from EHR prior to diagnosis of AD. They used natural language processing (NLP) from a list of 122 key words (grouped by category). They found that cases with AD had greater increase in relevant words compared to controls and that the AD group increased exponentially prior to diagnosis. These patterns were fairly similar by VA site and for stata of age, sex and race/ethnicity.

There is no “methods” section of the manuscript.

Response:

Thank you for pointing this out. The original submission followed the *Nature Aging* format, in which the Methods section is placed after the *Results and Discussion*. When the manuscript was transferred to *Communications Medicine*, we initially retained the original structure, which may have caused confusion and led to the perception that the Methods section was missing. In the current revision, we have reformatted the manuscript to align with *Communications Medicine*’s structure by moving the Methods section to follow the *Introduction* and precede the *Results*. We hope this resolves the misunderstanding and improves the clarity and accessibility of the manuscript.

Subjective cognitive decline (SCD) is a bit controversial as a precursor to AD as fairly nonspecific and often associated

with psychiatric and other medical conditions. The connection between mild cognitive impairment (MCI) is much stronger than for SCD. It is surprising they did not use MCI instead. The goal of using SCD was not clear really and distracting and not sure how it helps here.

Response:

Thank you for this valuable feedback. We agree that MCI is more strongly associated with future AD and plays a well-recognized role in the disease continuum. However, our study was designed to identify subtle, early indicators of cognitive decline as they naturally emerge in routine clinical care, often based on concerns raised by patients, family members, or clinicians—prior to the point at which a formal cognitive assessment is triggered.

Although our EHR data do include mentions of cognitive assessments (e.g., MMSE, MoCA), we purposefully excluded these mentions and related test results from our model inputs. This was done to avoid introducing bias from provider behavior (i.e., testing only when they already suspect cognitive decline), and to focus on symptom-based signals that may serve as passive, early flags of future AD risk. These real-world indicators are often aligned with the construct of SCD.

We acknowledge the limitations of SCD, but also recognize its growing relevance in the literature as a precursor to objective cognitive impairment, including its inclusion in Stage 2 of Clinical staging for individuals on the Alzheimer's disease continuum in the 2024 Revised criteria for diagnosis and staging of Alzheimer's disease by Jack et al.(2024). We have revised the manuscript to clarify our rationale for this choice.

Things really differed (exponentially) around 2 years prior to diagnosis (figure 3). This is interesting and it might make sense to look more closely at this time point.

Response:

Thank you for this insightful observation. We agree that the inflection around two years prior to diagnosis is both striking and clinically meaningful. In response, we conducted additional analyses to examine this turning point more closely, focusing on both SCD and AD dementia-related keyword frequencies and note volume trends across Year 3, Year 2, and Year 1 before diagnosis.

First, we disaggregated AD-related keyword frequencies by symptom category. As shown in Supplementary Figure 3(a), while all categories increased over time, neuropsychiatric symptoms exhibited the steepest increase in keyword counts between Year 2 and Year 1, indicating a notable acceleration in the documentation of behavioral and psychological concerns in the final year before diagnosis. Physiological changes also rose during this period. Learning/perception and memory-related terms showed smaller but consistent gains, suggesting broader symptom emergence approaching the diagnostic threshold.

We also analyzed note counts by specialty to identify which types of clinics patients most frequently visited in the years approaching diagnosis. The plot in Supplementary Figure 3(b) shows that certain specialties—particularly

geriatrics and cognitive care—had disproportionately higher increases in note volume between Year 2 and Year 1, consistent with rising care complexity.

We have added these findings to the revised manuscript, and we appreciate the reviewer’s suggestion, which helped us clarify and contextualize this key inflection point.

Supplementary Figure 3 (a). SCD and AD dementia–related keyword counts by category over the 3 years preceding diagnosis

Supplementary Figure 3 (b). Note volume by type in the 3 years preceding diagnosis

Along those lines, it is unclear what -1 day means and why choose 10 years before and then 1 day before diagnosis if that is what it means. The results are strikingly different. 1 day prior to diagnosis is unlikely to be that helpful for prediction and the AUC is much higher. I would like to see the results at 2 years prior as that seems to be a real inflection point but that was not investigated (see Figure 4).

Response:

Thank you for this thoughtful comment. We agree that -1 day prior to diagnosis does not reflect “early” prediction in a clinical sense, but we included it to represent the upper limit of our prediction model performance. In contrast, the -10 year time point captures the longest feasible prediction horizon given our dataset. Together, these two endpoints help define the model’s performance boundaries. Previous work by Tang et al. (2024) also reported similar time points in their manuscript (-1-day, -7-year).

We acknowledge the importance of intermediate time points—particularly the 2-year mark, which corresponds to the inflection point observed in the longitudinal keyword trajectories. In response, we have now added 2-year prediction results to the revised version of Table 2, alongside the previously reported results for -1, -3, -5, and -7 years.

We hope these additions improve the clarity and completeness of our study and better capture the trajectory of model performance over time.

The 10-year data is of interest but pretty low AUC. I think emphasizing the few years prior to diagnosis is more important. The 1-day prior to diagnosis had much higher AUC but then much less helpful prognostically.

Response:

Thank you for this important observation. We agree that the low AUC at 10 years reflects the inherent difficulty of predicting AD a decade in advance—especially when relying solely on early, subtle symptom documentation in unstructured clinical notes. This result illustrates the challenge of long-horizon prediction and helps define the lower bound of model performance in our study.

Conversely, while the model achieves much higher AUC at -1 day, we agree that this is less informative from a prognostic or early intervention standpoint, as it likely reflects symptomatology that has already triggered diagnostic action. We included it to represent the upper limit of our prediction model performance.

To provide a more clinically meaningful view of model performance, we also conducted experiments at multiple intermediate time points between -10 years and -1 day, including -7, -5, -3, -2, and -1 years prior to diagnosis. These results are included in Table 2, but in response to the reviewer's suggestion, we have revised the manuscript to better emphasize these intermediate time points in the main text rather than focusing solely on the extremes. This revised framing highlights the period where the model is most useful for early detection and potential intervention.

We appreciate the reviewer's comment, which has helped improve the clarity and clinical relevance of our findings.

Interestingly, the physiological changes and neuropsychiatric symptoms are the most noted early and then memory changes closer to diagnosis. ADLs are also big driver. It is important to reference some prior work by L Cleret de Langavant who used ML to identify dementia cases from EHR and other survey data. This idea of non-cognitive symptoms being useful early on is a key message and others have reported on this.

Response:

We appreciate the reviewer's insightful observation and fully agree that the temporal pattern of symptom emergence—where physiological and neuropsychiatric changes appear earlier, followed by memory-related complaints—is a key finding. Our stratified analysis by symptom category supports this sequence and highlights the potential utility of non-cognitive symptoms as early indicators of AD risk. We also agree that impairments in ADLs are a significant driver and align with emerging literature on functional decline as a predictive signal.

In response to the reviewer's helpful suggestion, we have now cited the work of L. Cleret de Langavant et al., which demonstrated the value of non-cognitive and behavioral features for early dementia identification using machine learning on EHR and survey data, along with other relevant studies. We have revised the Discussion to acknowledge this important body of work and to situate our findings within the broader context of research emphasizing non-cognitive prodromal markers.

Thank you for this valuable recommendation, which has strengthened the interpretability of our results.

Revision in manuscript:

Our stratified analysis revealed that physiological and neuropsychiatric symptoms—such as sleep disturbances, mood changes, and behavioral alterations—tended to appear earlier in the clinical record than explicit memory complaints, which became more prominent closer to diagnosis. Additionally, impairments in activities of daily living (ADLs) emerged as strong predictive features, consistent with the progressive functional decline characteristic of Alzheimer’s disease. These findings align with and extend prior work⁴⁶ used machine learning to detect dementia from EHR and survey data, emphasizing the predictive value of non-cognitive features in early stages of the disease. Our results support the growing recognition that non-cognitive symptoms may serve as valuable early indicators of AD risk,^{47–49} particularly when structured testing is not yet initiated. Leveraging such symptoms from routine clinical narratives could enhance early detection strategies and complement traditional memory-focused assessments.

Please elaborate on differences by clinic/specialty. There is one figure referring to primary care only.

Author Response:

Thank you for this insightful comment. Our primary goal is to enable early identification of AD, ideally years before formal diagnosis. We therefore focused on primary care settings, where most patients receive routine care and where early cognitive symptoms—such as those associated with SCD and AD dementia—are often first documented, even before specialist referral.

While our initial focus was on primary care notes, we agree that analyzing differences across specialties offers valuable insight. In response, we expanded our analysis to compare keyword frequencies across specialty groups.

Due to data sparsity in certain note types, we grouped specialties into broader, clinically relevant categories to ensure statistical reliability. These include:

- Primary
- Emergent Care (emergency visits)
- Mental Health (psychiatry, psychology, mental health clinics)
- Cognitive Specialty (memory clinics, neurology, neuropsychology, cognitive care)
- Geriatric Services (geriatrics, home-based primary care)
- Consultation Services (consults, compensation & pension examinations)

Supplementary Figure 1(f) shows the average counts of SCD and AD dementia–related keywords per patient per year for AD cases and matched controls, stratified by note types and years prior to diagnosis. Across all note types, SCD and AD dementia -related keyword counts all show a rapid increase in the years leading up to diagnosis for AD cases.

Mental health and geriatric services exhibited higher average keyword counts and the steepest increases in the final 2–3 years before diagnosis, highlight their central role in both documenting and managing advancing cognitive and behavioral symptoms along the trajectory to AD diagnosis.

Primary care notes, by contrast, exhibited a relatively high starting point and a gradual, consistent increase from as early as 15 years prior to diagnosis (Figure 3b), suggesting their importance in capturing early, longitudinal signs of cognitive decline. Cognitive specialty and emergent care notes showed lower overall counts but followed a similar upward trajectory approaching diagnosis.

These trends underscore the complementary contributions of generalist and specialist care settings in documenting the clinical trajectory toward AD.

Supplementary Figure 2 shows the distribution of note types by specialty in the case and control cohorts.

We then conducted experiments using those specialty note types individually for prediction, Supplementary Table 5 presents the results. Mental Health and Cognitive Specialty notes consistently demonstrated stronger predictive performance across timepoints.

We have incorporated these results and comparative analyses into the revised manuscript and Supplementary Materials. We appreciate the reviewer’s thoughtful suggestion to explore this dimension and believe it has strengthened the manuscript.

Supplementary Figure 1(f). Average SCD and AD dementia-related keywords per patient by year before diagnosis based on note types.

Supplementary Figure 2. Distribution of note types by specialty in the case and control cohorts in CP-I cohort.

Supplementary Table 5. Random forest prediction results using keyword features from different specialty note types in Setting I on the CP-I cohort.

	Emergent Care		Mental Health		Cognitive Specialty		Consultation Services		Geriatric Service	
	AUR OC	AUR OC	AUP RC	AUP RC	AUP RC	AUP RC	AUP RC	AUP RC	AUP RC	AUP RC
-1 day	0.618	0.384	0.648	0.455	0.642	0.447	0.561	0.311	0.603	0.385
-1 year	0.585	0.33	0.603	0.381	0.595	0.371	0.546	0.28	0.555	0.309
-2 year	0.567	0.305	0.583	0.35	0.575	0.338	0.539	0.265	0.539	0.283
-3 year	0.557	0.289	0.572	0.325	0.558	0.312	0.537	0.258	0.529	0.265
-5 year	0.547	0.269	0.549	0.289	0.538	0.278	0.527	0.248	0.517	0.247
-7 year	0.535	0.26	0.536	0.272	0.525	0.261	0.519	0.241	0.513	0.242

-10 year	0.519	0.304	0.522	0.314	0.511	0.301	0.510	0.236	0.504	0.292
----------	-------	-------	-------	-------	-------	-------	-------	-------	-------	-------

Figure 3 is a bit redundant...some can be put in supplement or mentioned in text. There are way too many figures anyway.

Response

Thank you for this helpful suggestion. We agree that Figure 3 contains multiple subpanels and that some visualizations may be better suited for the Supplement. In the revised manuscript, we have retained the most essential panels in the main figure to illustrate key trends (e.g., overall trajectory and primary care patterns), and moved the more granular stratified analyses (e.g., by race/ethnicity and keyword category) to the Supplementary Materials. We have also streamlined other figures to reduce redundancy and have updated the main text to clearly reference these supplementary figures where relevant. We appreciate the reviewer's feedback, which helped improve the clarity and focus of our visual presentation.